# Germline stem cell integrity and quiescence are controlled by an AMPK-dependent neuronal trafficking pathway

**Christopher Wong, Pratik Kadekar, Elena Jurczak, Richard Roy** *

Department of Biology, McGill University, Montreal, Canada

* richard.roy@mcgill.ca

**Data Availability Statement:** All relevant data are within the manuscript and its Supporting Information files.

**Funding:** This work was funded by a Project Grant (CIHR PJT-180267) awarded to RR by the

## Abstract

During periods of energetic stress, *Caenorhabditis elegans* can execute a developmentally quiescent stage called "dauer", during which all germline stem cells undergo a G2 cell cycle arrest. In animals that lack AMP-activated protein kinase (AMPK) signalling, the germ cells fail to arrest, undergo uncontrolled proliferation, and lose their reproductive capacity upon recovery from this quiescent stage. These germline defects are accompanied by, and likely result from, an altered chromatin landscape and gene expression program. Through genetic analysis we identified an allele of *tbc-7*, a predicted RabGAP protein that functions in the neurons, which when compromised, suppresses the germline hyperplasia in the dauer larvae, as well as the post-dauer sterility and somatic defects characteristic of AMPK mutants. This mutation also corrects the abundance and aberrant distribution of transcriptionally activating and repressive chromatin marks in animals that otherwise lack all AMPK signalling. We identified RAB-7 as one of the potential RAB proteins that is modulated by *tbc-7* and show that the activity of RAB-7 is critical for the maintenance of germ cell integrity during the dauer stage. We reveal that TBC-7 is regulated by AMPK through two mechanisms when the animals enter the dauer stage. Acutely, the AMPK-mediated phosphorylation of TBC-7 reduces its activity, potentially by autoinhibition, thereby preventing the inactivation of RAB-7. In the more long term, AMPK regulates the miRNAs *mir-1* and *mir-44* to attenuate *tbc-7* expression. Consistent with this, animals lacking *mir-1* and *mir-44* are post-dauer sterile, phenocopying the germline defects of AMPK mutants. Altogether, we have uncovered an AMPK-dependent and microRNA-regulated cellular trafficking pathway that is initiated in the neurons, and is critical to control germline gene expression cell non-autonomously in response to adverse environmental conditions.

## Author summary

During environmental challenges many organisms possess the ability to forego reproduction temporarily to enter a diapause stage only to resume reproduction when growing conditions improve. In *C. elegans*, the dauer stage is associated with a general developmental quiescence that is essential to preserve the reproductive competence of the germ

Canadian Institutes of Health Research (CIHR). The funders had no role in study design, data collection and analysis, decision to publish, or preparation of the manuscript.

**Competing interests:** The authors have declared that no competing interests exist.

cells following recovery, and this is controlled by AMPK signalling. However, we show that AMPK activation need not occur in all cells, but is critical in those cells that are likely most sensitive to such changes, namely in the neurons. Using genetic analysis, we revealed that the activation of AMPK in the neurons engages changes in RAB-7 trafficking, which are the result of a two-pronged AMPK-dependent regulation of a predicted RabGAP protein called TBC-7. By first modulating the activity of TBC-7, AMPK enhances RAB-7 activation, while later in the diapause, AMPK promotes the activity of two microRNAs that impinge on the *tbc-7* transcript, thereby blocking its expression. This results in a cell non-autonomous pro-quiescent signal that instructs the germ cells to modify their chromatin landscape and associated gene expression, ensuring that the germ cells remain reproductively competent for the duration of the diapause stage.

## Introduction

For many organisms, development unravels as a successive series of growth, quiescence, and phases of cell differentiation that proceed according to an organism-specific program. This is particularly true for closed systems, such as during embryonic development, where events often unfold independently of external resource availability due to maternal provisioning for the developing organism.

This situation changes dramatically as animals emerge from the embryo, and juveniles are subject to dramatic fluctuations in growth conditions. These challenges can be detrimental to continuous development and thus have driven the emergence of diverse mechanisms of adaptation that have evolved to enhance survival. One common means to circumvent these challenges is the programmed ability to undergo a period of developmental quiescence, or a diapause. In microbial cells, like bacteria and yeast, rapid growth and divisions arrest when resources are exhausted, while cells remain viable and poised to resume proliferation when growth conditions improve [1–3].

But this kind of quiescence is not unique to microbes. Mammalian cells in culture arrest their cell divisions and enter a $G_0$ state if they are deprived of growth factors (serum), glucose, or when they become confluent [4]. Moreover, cells that no longer appropriately execute quiescence in response to these cues can often demonstrate other rogue behaviours typical of transformed cells [3]. It is therefore highly advantageous to appropriately respond to these adverse environmental signals and execute a quiescent state as an adaptive means of protecting the cell and its invaluable genetic material.

Multicellular organisms also employ various states of quiescence to ensure survival during periods of environmental stress [5–7]. However, not all cells in the organism are capable of sensing these cues and therefore depend on specialized cells to communicate information to their distant neighbours to protect them against these challenges.

These specialized cells often express molecular sensors that are capable of both gauging external fluctuations and initiating the appropriate adaptive cellular/organismal changes to enhance survival during difficult environmental circumstances. Among the most widely studied of these cellular/molecular sensors is the stress-responsive AMP-activated protein kinase (AMPK). During periods of stress, it phosphorylates critical targets to block energy-consuming anabolic processes and activate pathways that will enhance the cellular energy pool. For example, many animals, including humans, forego reproduction during periods of excess stress or starvation [5,8]. Both starved *C. elegans* or mutants that lack insulin signalling when exposed to high temperatures during the L4 larval or early adult stages are infertile and do not

produce oocytes [9]. This is presumably because gametogenesis is very energy demanding and the resulting progeny would not fare well in a resource-depleted or unfavourable environment [9]. However, this oogenesis defect can be bypassed in these mutants by eliminating AMPK activity [8,10].

Activation of AMPK in *C. elegans* larvae that execute the highly-resistant, diapause-like dauer stage results in germline quiescence, while other somatic cell divisions are arrested through a parallel pathway [10,11]. In the absence of AMPK signalling, mutant germ cells continue to proliferate despite a lack of resources, resulting in sterility following recovery [8,10].

Animals can survive for months in the non-feeding dauer stage, yet when animals exit this diapause they are fully fertile with little or no negative reproductive consequence [12,13]. On the other hand, mutants that lack all AMPK signalling die prematurely during the dauer stage, and if they do recover they exhibit highly penetrant sterility [10]. The AMPK-mediated quiescence that occurs during the dauer stage is therefore protective and is critical to preserve germ cell integrity during this period of prolonged stress. Consistent with this, wild-type animals that transit through the dauer stage exhibit differences in their gene expression compared to animals that never encountered the stresses associated with dauer development [14,15]. These changes persist in the post-dauer animals as a molecular memory stored in the form of epigenetic marks, which have been shown to influence post-dauer fertility.

Mutants that lack all AMPK signalling display significant changes in gene expression during the dauer stage and as post-dauer adult animals when compared to wild-type animals [10]. In addition, the levels of transcriptionally activating and repressive chromatin marks within the germ line are abnormally upregulated, while their distribution is also abnormal. This suggests that the differences in gene expression in AMPK mutants could be attributed to the misregulation of chromatin marks in the affected germ cells.

Recent studies have indicated that AMPK can act cell non-autonomously to regulate life span [16], while the restoration of AMPK in some somatic cells was sufficient to correct the germline defects in AMPK mutants [10,17]. Namely, driving AMPK activity in the neurons and the excretory system was shown to suppress the AMPK germline defects [10]. Moreover, the post-dauer sterility and the germline hyperplasia typical of AMPK mutants are also partially suppressed when components of the endogenous small RNA pathway are compromised [10]. These findings suggest that several different pathways are likely to be involved in conveying the necessary signals to execute an adaptive gene expression response in the germ line in response to dauer entry. This is most likely coordinated through the function of specialized sensing cells; a role very often fulfilled by neurons. In the context of dauer development, AMPK would activate these various events in response to the metabolic challenges associated with dauer development in order to protect the germ line throughout the duration of the diapause.

Here we describe how AMPK regulates germ cell homeostasis in response to energy stress. This role requires its activity not in the germ cells, but in the neurons. By performing a genetic screen designed to identify genes involved in the AMPK regulation of germ cell integrity and germline quiescence during the dauer stage, we isolated eight alleles that partially restore germ cell function in these mutants. One of these genes encodes a RabGAP that enhances the intrinsic GTP hydrolysis activity of one or more Rab proteins, converting it from its active RAB-GTP form to its inactivated RAB-GDP form [18]. This RabGAP functions in the neurons to negatively regulate its RAB target. Furthermore, we demonstrate that the RabGAP is subject to regulation both directly by AMPK-mediated phosphorylation in the short term, and then in later stages of the diapause, by regulating its expression by two microRNAs. Therefore, through the successive targeting of a single RabGAP in the neurons, AMPK signals cell non-autonomously to the germline stem cells to arrest their cell divisions and to maintain germ cell integrity at the onset of this period of developmental quiescence.

## Results

### Mutations that reverse the germline defects of AMPK mutants

The *C. elegans* dauer stage is marked by a pervasive organismal developmental quiescence that is initiated as animals transit through the preparatory L2d stage. This is accompanied by a complete arrest in germ cell divisions mediated by the activity of the *C. elegans* orthologues of the human tumour suppressor genes PTEN (*daf-18*), LKB1 (*par-4*), and one of its many downstream targets, AMPK [8]. Animals that lack all AMPK signalling due to the deletion of both catalytic subunits *aak-1* and *aak-2* (referred to as *aak(0)*) undergo extensive germline hyperplasia during dauer formation and exhibit post-dauer sterility, in addition to a number of metabolic/somatic phenotypes [10,17]. Curiously, the compromise of various components involved in the biogenesis and function of small RNAs can partially restore germ cell quiescence in AMPK mutant dauer larvae, in addition to the corresponding post-dauer fertility in these mutants.

To identify additional genes that converge either on these small RNA regulators or on AMPK targets in the germ line, we performed a forward genetic screen to obtain mutants that suppress the germline defects typical of AMPK mutants. Eight alleles were isolated that fall into five complementation groups (Table 1), each of which suppressed the germ cell hyperplasia and/or the post-dauer sterility in AMPK mutants to varying degrees (S1A–S1C Fig and S1 Table). All isolated alleles were able to lay viable embryos that gave rise to a fertile generation.

The *rr166* allele was the most effective in suppressing the AMPK germline and somatic defects (Fig 1A–1C). When AMPK mutants recover from the dauer stage, many animals die prematurely. Those mutants that recover display diverse abnormalities in vulval development, such as protruding vulva, burst vulva, or multivulva [10]. In addition to its effects on the germline hyperplasia, *rr166* suppressed most of the post-dauer somatic defects. This allele may therefore not be exclusively active in the germ line and could have a function in the soma during normal development and/or in the dauer stage. Alternatively, it could adjust cell divisions in somatic tissues through its effect on the germ line (S1 Table).

**Table 1. Mutants that suppress the sterility of post-dauer AMPK mutants fall into five complementation groups.** Eight EMS-generated alleles were isolated from a genetic screen designed to identify genes involved in the AMPK regulation of germ cell integrity and germline quiescence during the dauer stage. All eight alleles were able to lay viable embryos that gave rise to a fertile generation. Through cross complementation analysis they were found to fall into five complementation groups, two of which had multiple independent alleles; Group 1—*rr166*, *rr267* and Group 3—*rr256*, *rr266*, *rr289*. Based on our $F_1$ and $F_2$ cross-complementation analysis, all the alleles behave recessively, with the exception of *rr249*, which acts semi-dominantly. Wild type-like indicates the lack of any post-dauer somatic phenotypes associated with the loss of AMPK signalling, such as protruding vulva, burst vulva, multivulva, or premature death during the recovery period from the dauer stage. The values for % egg laying post-dauer animals, no. of germ cells in dauer larvae, and % wild type-like are presented as means. All strains contain the *daf-2(e1370)* allele. All EMS mutant strains are *daf-2; aak(0)*. Each assay was repeated three times with 50 animals in each trial. n = 50.

| Allele(s) | Complementation Group | Dominance/ Recessiveness | Penetrance | | |
|---|---|---|---|---|---|
| | | | % egg laying post-dauer animals | No. of germ cells in dauer larvae | % wild type-like |
| *daf-2* | - | - | 100 | 35 | 100 |
| *daf-2; aak(0)* | - | - | 0 | 153 | 68 |
| *rr166* | Group 1 | Recessive | 83 | 68 | 92 |
| *rr249* | Group 2 | Semi-dominant | 78 | 97 | 84 |
| *rr256* | Group 3 | Recessive | 67 | 85 | 84 |
| *rr261* | Group 4 | Recessive | 67 | 90 | 76 |
| *rr266* | Group 3 | Recessive | 63 | 66 | 84 |
| *rr267* | Group 1 | Recessive | 67 | 97 | 80 |
| *rr268* | Group 5 | Recessive | 68 | 75 | 76 |
| *rr289* | Group 3 | Recessive | 61 | 112 | 72 |

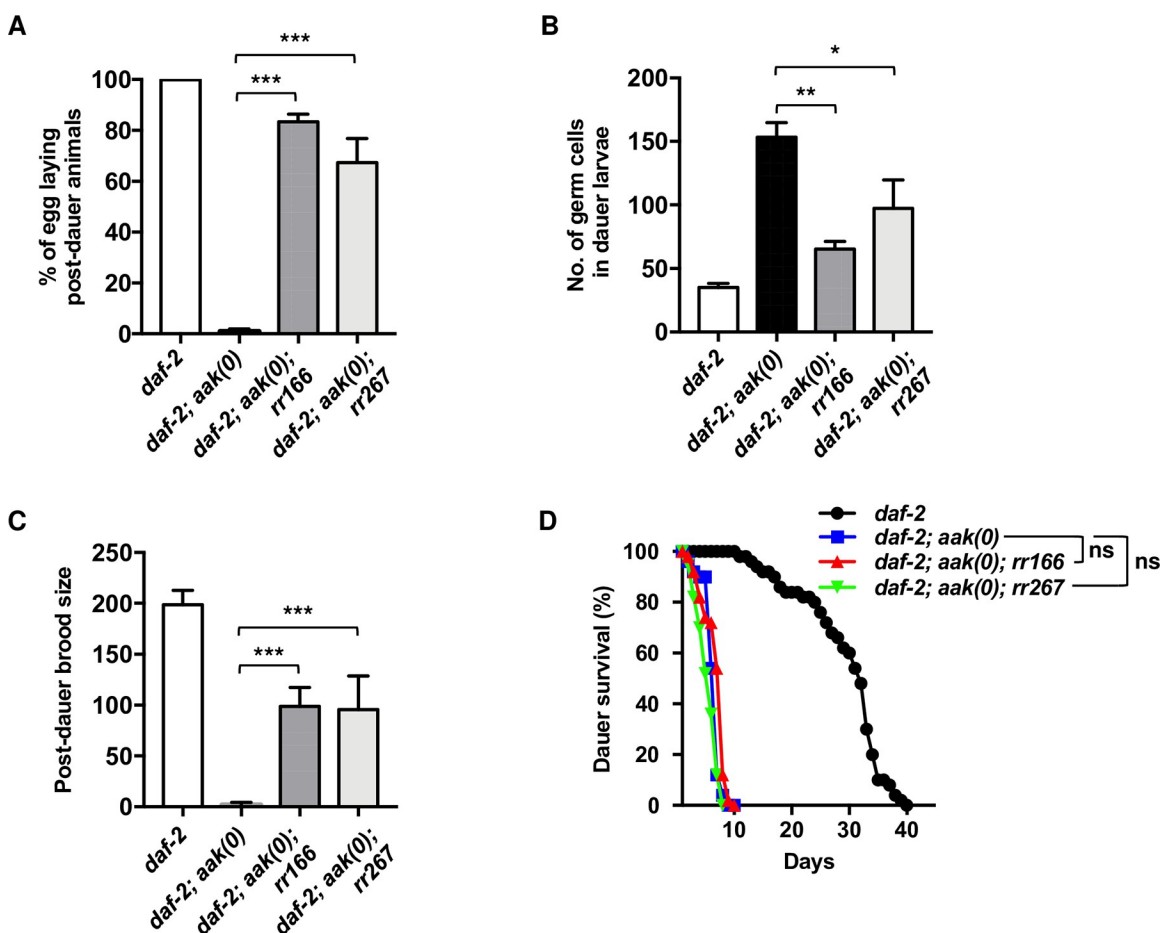

**Fig 1. *rr166* and *rr267* partially suppress the post-dauer sterility and dauer germline hyperplasia of AMPK mutants.** (A-C) *rr166* and *rr267* suppress the (A) post-dauer sterility, (B) dauer germline hyperplasia, and (C) brood size defects in animals that lack all AMPK signalling. (D) *rr166* and *rr267* do not suppress the reduced dauer survival typical of *aak(0)* mutants. ***P < 0.0001 when compared to *daf-2; aak(0)* using ordinary one-way ANOVA for post-dauer brood size and no. of germ cells in dauer larvae. Data are represented as mean ± SD. ***P < 0.0001 when compared to *daf-2; aak(0)* using Marascuilo procedure for % egg laying post-dauer animals (% total). No significant difference (ns) was observed when compared to *daf-2; aak(0)* using a log-rank test for dauer survival. All animals carry the *daf-2(e1370)* allele. The values for % egg laying post-dauer animals, no. of germ cells in dauer larvae, and post-dauer brood size are presented as means. Each assay was repeated three times with 50 animals in each trial. n = 50.

AMPK is also required for the typical long-term survival of the dauer larva by regulating tri-glyceride hydrolysis during the dauer stage [17]. Dauer larvae with intact AMPK signalling can survive for several weeks/months, while mutants with disrupted AMPK/LKB1 signalling die prematurely due to the inappropriate depletion of stored lipids important to sustain animals. Strikingly, neither *rr166* nor *rr267* improved the dauer survival of *aak(0)* mutants, suggesting that the *rr166* and *rr267* mutations do not affect other AMPK-regulated metabolic pathways and may be more specific to germ line/gonadal physiology (Fig 1D).

### *rr166* corrects the abnormal deposition of the chromatin marks in the germ line of AMPK mutant dauer larvae and post-dauer adults

*daf-2; aak(0)* mutants not only have upregulated levels of both transcriptionally activating and repressive chromatin marks, but they also have an abnormal distribution of these epigenetic modifications across the germ line in both dauer and post-dauer animals [10]. This altered

chromatin landscape is associated with a dramatic change in germline gene expression during the dauer stage, which remains unaltered upon dauer exit, persisting into the post-dauer adult. To better understand how *rr166* suppresses the AMPK germline defects, we examined the levels of chromatin marks that were aberrant in the AMPK mutant dauer larvae to assess how these epigenetic modifications are affected in the *daf-2; aak(0); rr166* mutant. Western analysis revealed that both activating (H3K4me3) and repressive (H3K9me3) chromatin modifications were restored to nearly normal levels in the germ cell nuclei of *daf-2; aak(0); rr166* dauer larvae that lack all AMPK signalling (S2A–S2A" Fig). In addition, the distribution of the chromatin marks was also corrected (S2B–S2C" Fig).

Using the same approach, we confirmed that the *rr166* mutation was also able to suppress the increased abundance of the chromatin marks in the *daf-2; aak(0)* mutants during the post-dauer stage (S3A–S3A" Fig). Immunofluorescence analysis against these histone modifications indicated that *rr166* could correct the abundance of each mark in individual nuclei, while also re-establishing the overall distribution pattern across the germ line (S3B–S3C" Fig). Interestingly, while the levels of H3K9me3 in the *rr166* mutants were similar compared to *daf-2* control animals, the levels of H3K4me3 were not completely restored to *daf-2* control levels, albeit they were reduced compared to those of *daf-2; aak(0)* mutants (S3B–S3B" Fig). Together, these data indicate that *rr166* corrects the abundance and distribution of chromatin marks in the germ line that presumably prime germline gene expression for a period of developmental quiescence, in animals that lack all AMPK signalling.

## Misregulated RabGAP activity in neurons of AMPK mutants results in germline abnormalities during the dauer stage

To identify relevant polymorphisms that could correspond to the affected gene responsible for the suppression of AMPK germline defects, genomic DNA from *daf-2; aak(0); rr166* mutants was subjected to next generation sequence analysis [19]. *rr166* was revealed to be a typical EMS-generated G->A transition mutation in a gene called *tbc-7* that encodes a predicted Rab-GAP protein.

To confirm that *rr166* is indeed an allele of *tbc-7*, we performed RNAi against *tbc-7* in the *daf-2; aak(0)* mutants. Fertility was partially restored in *daf-2; aak(0); tbc-7* (RNAi) animals, and they showed no significant difference in their post-dauer fertility when compared to *daf-2; aak(0); rr166* animals (Fig 2A). Furthermore, when *daf-2; aak(0); rr166* mutants were injected with a fosmid which contained a wild-type copy of *tbc-7*, the suppression of the AMPK mutant germline defects was reverted, suggesting that *rr166* is indeed an allele of *tbc-7* (Fig 2A).

The *rr166* allele is a missense mutation that results in a change from serine to phenylalanine at position 520, which corresponds to the predicted TLDc domain of *tbc-7*. From the same complementation group, the *rr267* allele corresponds to a missense mutation that alters a phenylalanine to leucine at position 156, which does not correspond to any predicted protein domain. The difference in the position of the mutations could account for the ability of each allele to suppress the AMPK germline defects; compromising the TLDc domain with the S520F mutation could have a greater impact on TBC-7 function compared to the F156L mutation.

Mutations that eliminate the entire *tbc-7* region are presumably non-viable, as no mutants that contain a large or complete deletion of *tbc-7* can be maintained as a homozygous animal, but *rr166* is viable. To determine if the *tbc-7(rr166)* point mutation is a hypomorphic allele of an essential gene, we crossed the *tbc-7(rr166)* mutation into a null mutant that contains a deletion of the entire *tbc-7* gene (*tm10766*). Our subsequent genetic analyses were consistent with

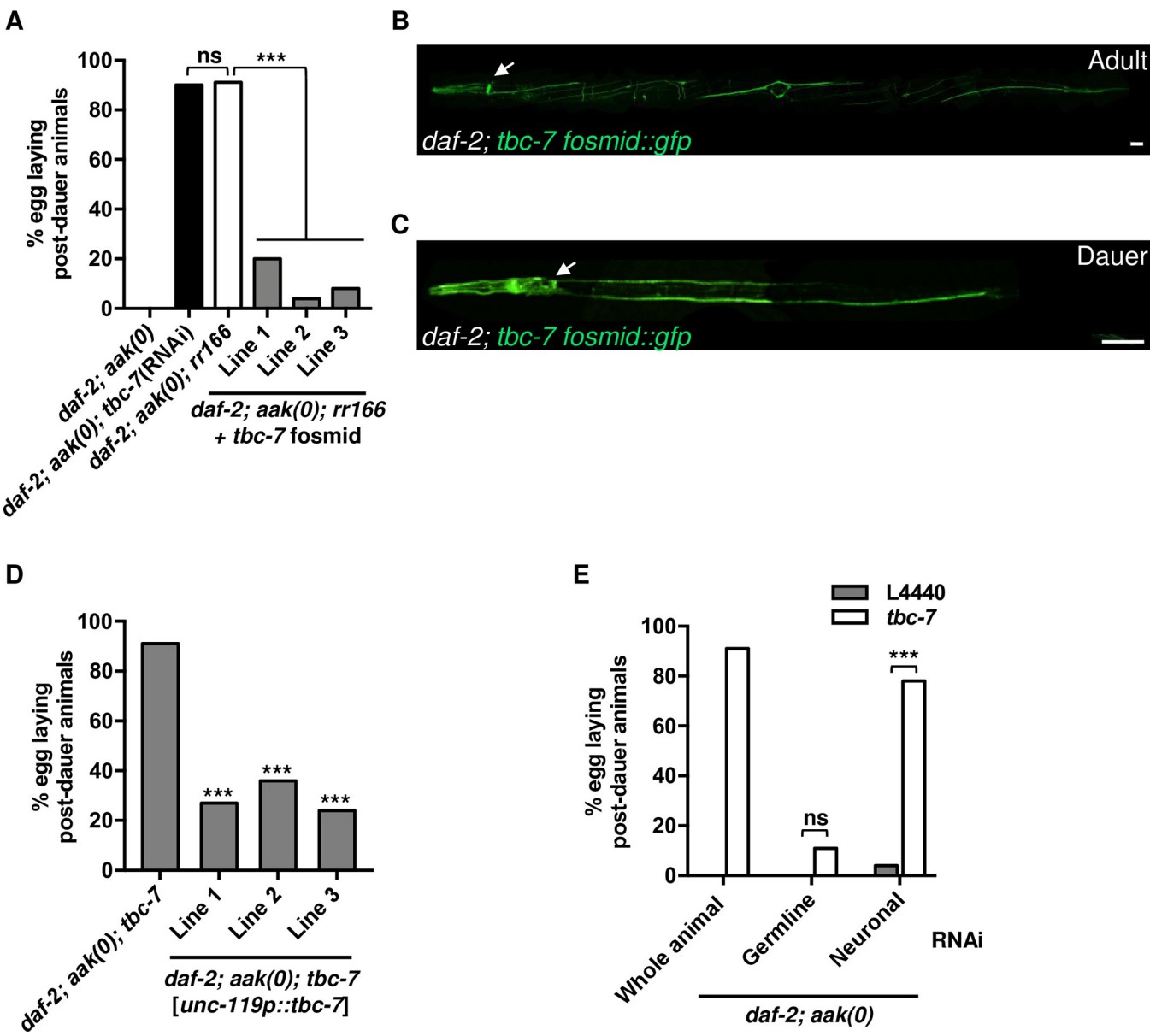

**Fig 2. *tbc-7* functions in the neurons to maintain germ cell integrity during the dauer stage.** (A) *rr166* is an allele of the RabGAP *tbc-7*. RNAi against *tbc-7* in *daf-2; aak(0)* suppresses the post-dauer sterility. Injection of a fosmid containing a wild-type copy of *tbc-7* (C31H2.1) into *daf-2; aak(0); tbc-7* mutants partially reverts the *tbc-7*-associated suppression of the *aak(0)* germline defects. ***P < 0.0001 when comparing *daf-2; aak(0); rr166* and *daf-2; aak(0); rr166* + *tbc-7* fosmid based on Marascuilo procedure for % egg laying post-dauer animals. ns when comparing *daf-2; aak(0); tbc-7(RNAi)* and *daf-2; aak(0); rr166* based on Marascuilo procedure for % egg laying post-dauer animals. (B-C) The expression of a fosmid containing *tbc-7* translationally fused to GFP is shown in (B) wild-type adult animals and in (C) dauer larvae. White arrows indicate the nerve ring. Scale bars: 25 μm. Note that the morphological abnormality visible in the image is a result of the *rol-6D* co-transformation marker. (D) A wild-type copy of *tbc-7* cDNA expressed exclusively in the neurons by the *unc-119* promoter in the *tbc-7*-suppressed mutants reverts the suppression of the AMPK germline phenotypes. ***P < 0.0001 when compared to *daf-2; aak(0); tbc-7* based on Marascuilo procedure for % egg laying post-dauer animals. (E) Tissue-specific RNAi experiments reveal that the compromise of *tbc-7* expression exclusively in the neurons (Neuronal) is sufficient to suppress the AMPK germline defects. ***P < 0.0001 when compared to L4440 empty vector based on Marascuilo procedure for % egg laying post-dauer animals. All animals carry the *daf-2(e1370)* allele. The values for % egg laying post-dauer animals are presented as means. Each assay was repeated three times with 50 animals in each trial. n = 50.

*rr166* acting as a recessive, hypomorphic allele of *tbc-7* (S4A and S4B Fig and Materials and Methods).

Recent work demonstrated that AMPK can regulate germ cell integrity cell non-autonomously during the dauer stage [10]. Restoring AMPK function in the neurons and in the

excretory system re-established germline quiescence and germ cell integrity in AMPK mutants [10]. Furthermore, the somatic expression of *aak-2* was able to correct both the abundance and distribution of chromatin marks in the dauer germ line, suggesting that the somatic function of AMPK controls the execution of germline quiescence in response to dauer cues.

*tbc-7* encodes a RabGAP protein that is conserved in *Drosophila* and in humans, where it is critical for appropriate synaptic vesicle dynamics in neurons [20]. Because of this neuronal role, we wondered if *tbc-7* might regulate the germ line cell non-autonomously during the dauer stage. To better understand where *tbc-7* is expressed we generated a transgenic strain using a fosmid containing a TBC-7::GFP translational fusion protein. This fosmid contains the entire coding sequence of *tbc-7* as well as the 5'UTR, 3'UTR, and any potential upstream and downstream regulatory sequences (starts X:5,120,813 and ends X:5,154,839) (See Materials and Methods). A previous report suggested that *tbc-7* might act predominantly in muscle to regulate autophagy, but this may not be entirely accurate, since this expression analysis was based on a relatively small region of upstream sequence used to drive a transcriptional fusion transgene [21]. Fosmid-based reporter systems contain large genomic regions that are more likely to capture most, if not all, cis-regulatory information, therefore resulting in a more accurate representation of the expression of any given gene [22].

Imaging revealed that TBC-7 is highly expressed in the neurons of adult animals grown in replete conditions as well as in dauer larvae (Fig 2B and 2C). Moreover, the neuronal expression of *tbc-7* is critical for its function in this context, since the introduction of a wild-type copy of *tbc-7* cDNA driven by a pan-neuronal promoter in the *daf-2; aak(0); tbc-7* mutants reverted the suppression of AMPK germline defects (Fig 2D). Because of the claim that *tbc-7* might also function in the body wall muscles [21], a wild-type copy of *tbc-7* cDNA was driven by a muscle-specific promoter (*myo-3*) in *daf-2; aak(0); tbc-7* mutants to determine if this tissue contributes to the regulation of dauer germ line integrity. Unlike our findings using a neuron-specific *tbc-7* transgene, driving the expression of *tbc-7* exclusively in the muscles was unable to revert the suppression of the germline defects, suggesting that *tbc-7* does not regulate germ cell integrity from the body wall muscle, and its role in antagonizing germ cell quiescence during the dauer stage is dependent on its neuronal function (S5A Fig).

Our initial findings indicated that when AMPK function is restored in the excretory system, or in the nervous system, the post-dauer sterility and the germline hyperplasia are corrected, suggesting that AMPK may also work in the excretory system to regulate germ cell integrity during the dauer stage [10]. To determine if *tbc-7* expression in the excretory system may also contribute to this AMPK effect on the germ line, a wild-type copy of *tbc-7* cDNA was driven by an excretory system-specific promoter (*sulp-5*) in *daf-2; aak(0); tbc-7* mutants. Similar to the muscle-specific expression, this excretory-specific transgene had no effect on the *tbc-7*-associated suppression of the AMPK mutant phenotypes. Therefore, AMPK must somehow modulate *tbc-7* function in the neurons during the onset of the dauer stage to instruct the germ line to execute quiescence (S5B Fig).

To further confirm that *tbc-7* acts in the neurons, we used a tissue-specific RNAi strategy that allowed us to compromise the expression of *tbc-7* exclusively in one tissue at a time (see Materials and Methods) (S2 Table) [23,24]. If *tbc-7* activity is required in the neurons, then *daf-2; aak(0)* mutants that are subjected to *tbc-7* RNAi exclusively in the neurons should phenocopy the *tbc-7(rr166)* mutation and suppress the AMPK germline defects. Using the neuronal RNAi strain, we found that *tbc-7* RNAi indeed suppressed the post-dauer sterility of *daf-2; aak(0)* mutants, while strains that had *tbc-7* compromised exclusively in the germ line showed no such suppression, although the RNAi was indeed effective (Figs 2E and S8). Taken together, these data indicate that TBC-7 functions in the neurons to antagonize AMPK-dependent germline quiescence and germ cell integrity during the dauer stage.

## TBC-7 negatively regulates RAB-7 in the neurons to maintain germ cell integrity

TBC-7 is a predicted RabGAP protein that could enhance the intrinsic GTP hydrolysis activity of Rab GTPases, presumably converting them from their active GTP-bound form into their inactive GDP-bound form. In AMPK mutants, the *tbc-7*-associated RAB(s) are most probably in their inactive GDP-bound form due to misregulated TBC-7 activity, while in *daf-2; aak(0); tbc-7* mutants, the *tbc-7*-associated RAB(s) may be in their active GTP-bound form (Fig 3A).

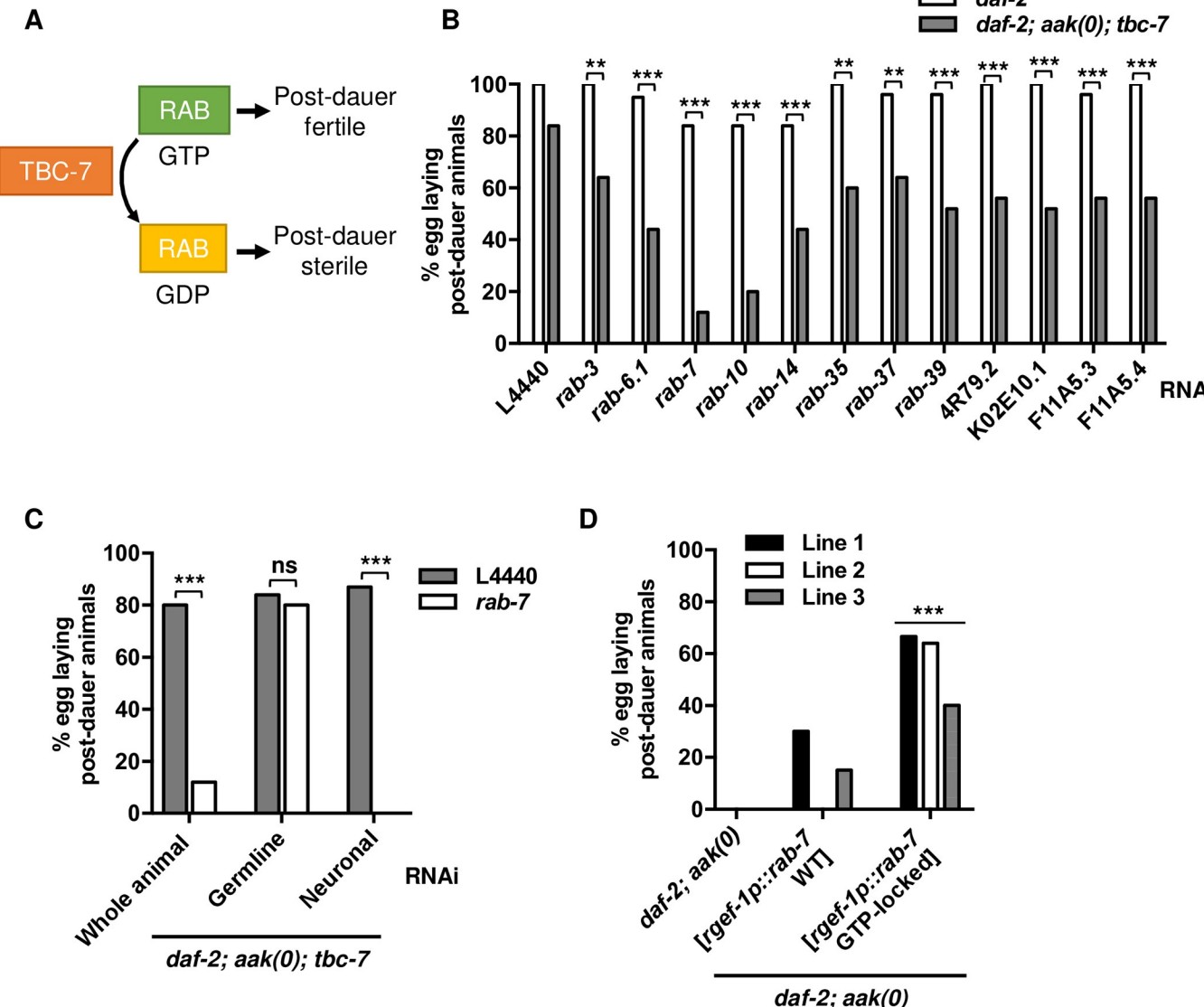

**Fig 3. TBC-7 regulates RAB-7 in the neurons to maintain germ cell integrity during the dauer stage.** (A) Graphical figure illustrating the regulation of RAB activity by TBC-7. In *daf-2; aak(0); tbc-7* mutants, TBC-7 is inactive, allowing its RAB protein to be active in its GTP-bound state. By compromising the expression of this RAB protein using RNAi, the suppression of post-dauer sterility is lost. (B) By subjecting each predicted known *rab* gene in the *C. elegans* genome to RNAi, several RAB proteins were identified as potential RAB partners of TBC-7. All of the RNAi treatments were performed in animals with the *daf-2(e1370)* background. ***P < 0.0001, **P < 0.001 when compared to *daf-2* with the same RNAi treatment using Marascuilo procedure for % of egg laying animals. (C) Tissue-specific RNAi experiments reveal that *rab-7* expression is required in the neurons and not the germ line for the *tbc-7* suppression of AMPK germline phenotypes. ***P < 0.0001, **P < 0.001 when compared to L4440 empty vector using Marascuilo procedure for % of egg laying animals. (D) Introduction of a GTP-locked variant of RAB-7 (Q68L) [26,27] into *daf-2; aak(0)* mutants partially suppresses the post-dauer sterility. ***P < 0.0001 when compared to *daf-2; aak(0)* mutants expressing neuronal wild-type *rab-7* using Marascuilo procedure for % of egg laying animals. All animals carry the *daf-2 (e1370)* allele. The values for % egg laying post-dauer animals are presented as means. Each assay was repeated three times with 50 animals in each trial. n = 50.

Therefore, compromising the expression of the RAB(s) associated with *tbc-7* should revert the suppression conferred by the *tbc-7* mutation, causing a loss of germ cell integrity and post-dauer fertility. To identify which RAB protein is regulated by *tbc-7*, each predicted *rab* gene in the *C. elegans* genome was compromised using a hypomorphic RNAi strategy [25]. This RNAi survey of RAB proteins revealed that TBC-7 may regulate a large number of RAB effectors (Fig 3B and S3 Table). However, when *rab-7* was compromised in the *daf-2; aak(0); tbc-7* mutants, the post-dauer fertility was reduced to a level similar to that of post-dauer *daf-2; aak(0)* mutants (compare the fourth column in Fig 3B with the first column in Fig 2A). Furthermore, if *rab-7* acts downstream of AMPK signalling and *tbc-7* to regulate germline quiescence in the dauer stage, then disabling it should result in dauer phenotypes very similar or identical to AMPK mutants. Indeed, *rab-7* RNAi in the *daf-2; aak(0); tbc-7* mutant during the dauer stage caused post-dauer sterility (Fig 3B, fourth column). Taken together, these data strongly support a role for *rab-7* in the correct regulation of germline quiescence during the dauer stage.

The compromise of *rab-10* in the *daf-2; aak(0); tbc-7* mutants showed a similar phenotype to the loss of *rab-7* through RNAi. Therefore, to examine if RAB-10 could be a direct target of TBC-7, we employed our tissue-specific RNAi system in the *daf-2; aak(0); tbc-7* mutant to examine in which tissue RAB-10 functions during the dauer stage. When *rab-10* was compromised exclusively in the germ line, the animals were post-dauer sterile, while the compromise in the neurons had no observable effects on the post-dauer fertility, suggesting that *rab-10* functions in the germ line (S6 Fig). We therefore reasoned that RAB-10 is most likely not a direct Rab target of TBC-7 in this context, since they function in different tissues.

To further confirm that RAB-7 is regulated by TBC-7 in the neurons, we once again used our tissue-specific RNAi system [23,24] to attenuate *rab-7* activity exclusively in the neurons, which resulted in highly penetrant post-dauer sterility (S4 Table), while its compromise in the germ line had no observable effects on the reproductive output of the post-dauer animal (Fig 3C).

Rab proteins bind GTP and then hydrolyse it to GDP through their intrinsic GTPase activity, which is greatly enhanced by the activity of its corresponding RabGAP [18]. By using a mutated form of a Rab protein in which the GTPase activity has been disabled (GTP-locked), a constitutively active form of the Rab can be used to assess the various roles of the activated Rab protein [26,27]. Using this strategy, we introduced a GTP-locked variant of RAB-7 in the neurons of the *daf-2; aak(0)* animals by introducing a point mutation Q68L in RAB-7 driven by the *rgef-1* neuronal promoter [26,28]. In mammalian cells, the GTP-hydrolysis deficient mutants Rab7 Q67L are locked in its GTP-bound form, resulting in constitutive activity [28]. The orthologous mutation in *C. elegans* RAB-7 Q68L results in a GTP-locked variant of RAB-7 that when expressed in the intestine, results in larger intestinal vesicles [26]. If RAB-7 is involved in establishing or maintaining germline quiescence and is misregulated by TBC-7 when AMPK is no longer functional, then the presence of a GTP-locked variant in this context should result in the suppression of the post-dauer sterility presumably caused by misregulated TBC-7 activity in the absence of AMPK signalling. Indeed, the expression of the GTP-locked RAB-7 variant improved post-dauer fertility (average of 56%) in the AMPK mutant dauer larvae (Fig 3D). Taken altogether, our analyses indicate that misregulated TBC-7 blocks RAB-7 activation in the neurons of dauer larvae, thereby allowing germ cell divisions to continue, eventually resulting in the germline hyperplasia typical of AMPK mutants.

## *mir-1* and *mir-44* negatively regulate *tbc-7* downstream of AMPK activation to promote germline quiescence

The RabGAP *tbc-7* has been previously reported to be regulated by *mir-1* [21,29], whereby *mir-1* binds directly to the 3'UTR of *tbc-7* to silence its expression. The loss of the *mir-1*

binding site or the loss of *mir-1* itself results in an overexpression of *tbc-7*, causing impaired autophagy and proteotoxic stress [21]. In a separate study that sequenced miRNA-target sites bound by ALG-1, *mir-1* was found to bind the 3'UTR of *tbc-7* directly, further suggesting that *mir-1* could be a direct regulator of *tbc-7* expression [30]. microRNA production is critical for entering the dauer stage and genetic mutants that are deficient in producing specific micro-RNAs, or do not possess the machinery necessary for microRNA production, are dauer defi-cient [31]. However, it is possible that in addition to their critical role in dauer formation, microRNAs may also be involved in regulating multiple key targets that are necessary for vari-ous aspects of the diapause, including the maintenance of germ cell integrity.

To determine if any potential small RNAs could impinge on *tbc-7* function, we analysed the predicted *tbc-7* transcript using TargetScanWorm (version 6.2), a program designed to mea-sure the biological relevance between predicted miRNAs and seed sequences. We identified two highly conserved seed sequences for *mir-1* and one highly conserved seed sequence for *mir-44* in the 3' UTR of *tbc-7* (S5 Table). *mir-1* is one of 32 conserved microRNAs throughout the Bilateria and is important for many functions including autophagy, muscle development, and sarcomere and mitochondrial integrity [32,33]. The *mir-44* family modulates the germline sex determination pathway in *C. elegans* by promoting the production of sperm in the her-maphrodite germ line [34].

If *mir-1* and *mir-44* regulate the expression of *tbc-7* directly, then the loss of either or both of them during the dauer stage should result in the misregulation of *tbc-7*. This increase in TBC-7 activity would further enhance the hydrolysis of RAB-7 GTP into RAB-7 GDP, result-ing in the same post-dauer phenotypes seen in *daf-2; aak(0)* mutants. Curiously, when mutants that contain deletions of the *mir-1* microRNA (*mir-1(gk276)*) and deletions of the entire *mir-44* microRNA family (*nDf49* allele: *mir-44, ZK930.12, mir-42, ZK930.15, mir-43*) were allowed to spend 24 hours in the dauer stage, there were no observable defects in the post-dauer fertil-ity (Fig 4B). However, if the duration of dauer is prolonged to 7 days, *mir-1; daf-2* and *mir-44; daf-2* mutants exhibited varying degrees of post-dauer sterility, while *daf-2* control animals were unaffected (Fig 4B). After allowing *mir-1; daf-2* and *mir-44; daf-2* mutants to recover after 7 days in the dauer stage, the mutants were found to recover at normal rates compared to *daf-2* control animals and AMPK mutants, but their fertility was greatly decreased, such that they resembled AMPK mutants. This finding suggests that the germ cells may lose their integ-rity after a prolonged duration in the dauer stage in the *mir-1; daf-2* and *mir-44; daf-2* mutants, presumably due to the inappropriate regulation of *tbc-7*. Furthermore, when the activity of the TBC-7 target RAB-7 was further reduced by performing *rab-7* RNAi, the *mir-1; daf-2* and *mir-44; daf-2* mutants exhibited post-dauer sterility after only 24 hours in the dauer stage (Fig 4C). Moreover, *mir-1; mir-44; daf-2* mutants exhibited similar post-dauer germline defects after spending 7 days in the dauer stage or after 24 hours after being subjected to *rab-7* RNAi (S7A and S7B Fig). This enhancement of the *mir-1* and *mir-44* post-dauer sterility following *rab-7* RNAi is consistent with a role of *mir-1* and *mir-44* in the regulation of *tbc-7*. Our findings sug-gest that as the animal spends an extended period of time in the dauer stage, the loss of *mir-1* and *mir-44* presumably allows for *tbc-7* transcripts to be actively expressed. TBC-7 can then continually enhance the hydrolysis of RAB-7 GTP into RAB-7 GDP, decreasing the neuronal pool of available active RAB-7. It is this depletion of active RAB-7 in its GTP-bound state that is most likely responsible for the observed germ cell hyperplasia and the loss of germ cell integ-rity, rendering the post-dauer animals sterile.

If *mir-1* and *mir-44* do indeed bind to the 3'UTR of *tbc-7* to negatively regulate its expres-sion during the dauer stage, then the deletion of these sequences should lead to increased *tbc-7* expression, thereby reducing the levels of GTP-bound RAB-7 and a consequent loss of post-dauer fertility. In *daf-2* control animals that harbour wild-type AMPK function, but possess a

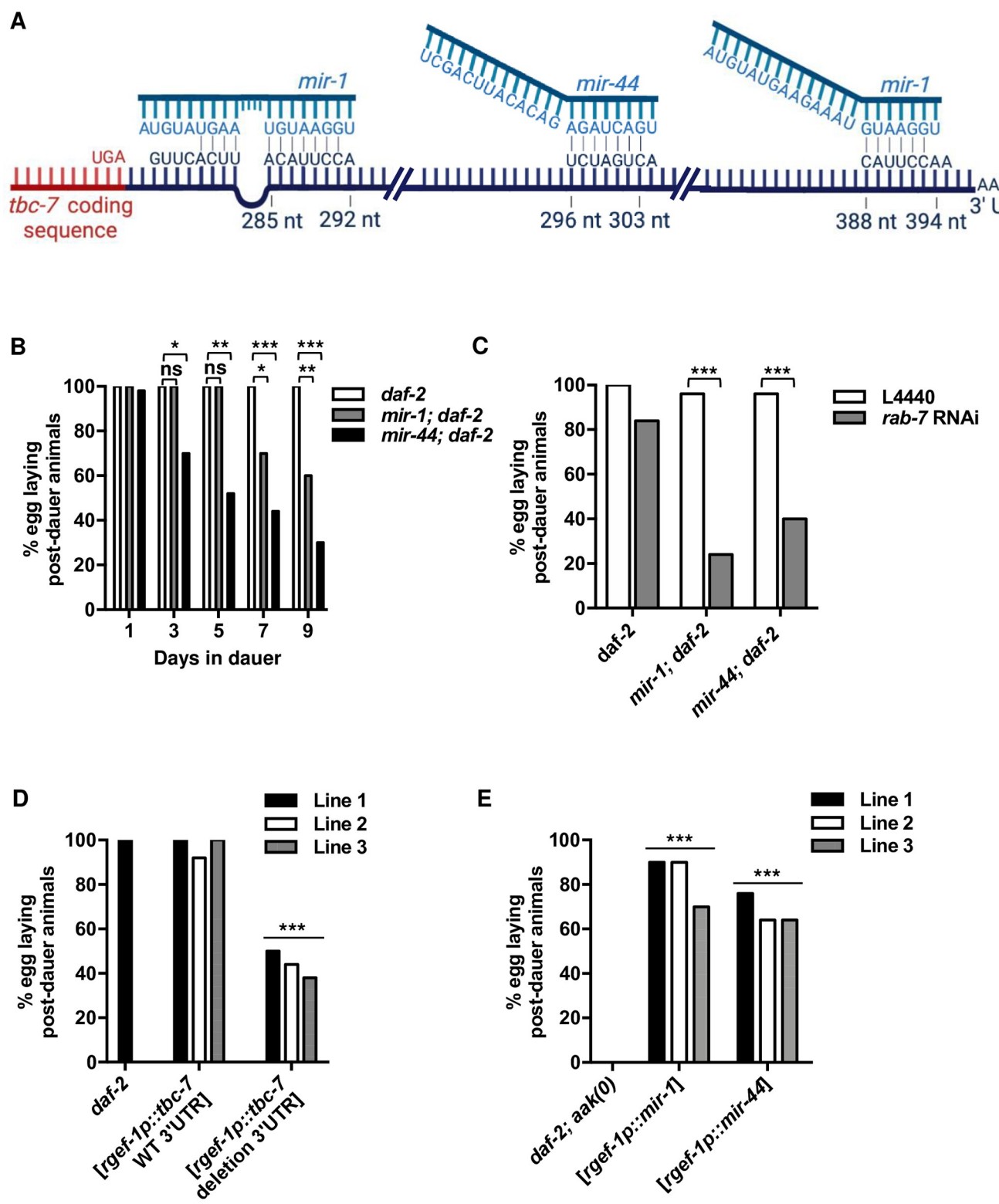

**Fig 4. *mir-1* and *mir-44* are required to maintain germ cell integrity during the dauer stage.** (A) A model illustrating the binding of *mir-1* to the two predicted seed sequences and *mir-44* to the one predicted seed sequence on the 3' UTR of the *tbc-7* transcript. Lines indicate predicted binding between the microRNAs and *tbc-7* 3' UTR. (B) *mir-1; daf-2* and *mir-44; daf-2* mutants exhibit post-dauer sterility after 7 days in the dauer stage. ***P < 0.0001,

**P < 0.001, *P < 0.01 when compared to *daf-2* that spent an identical duration in the dauer stage using Marascuilo procedure for % egg laying post-dauer animals. (C) Reducing *rab-7* levels greatly enhances/accelerates the post-dauer sterility associated with a loss of *mir-1* or *mir-44*. ***P < 0.0001 when compared to L4440 empty vector using Marascuilo procedure for % egg laying post-dauer animals. (D) Mutants with a deletion of the *mir-1* and *mir-44* seed sequence on the 3'UTR of *tbc-7* exhibit post-dauer sterility after 1 day in the dauer stage. ***P < 0.0001 when compared to *daf-2* expressing a wild-type *tbc-7* 3'UTR using Marascuilo procedure for % egg laying post-dauer animals. (E) Increased *mir-1* or *mir-44* expression in *daf-2; aak(0)* mutants suppresses post-dauer sterility. ***P < 0.0001 when compared to *daf-2; aak(0)* using Marascuilo procedure for % egg laying post-dauer animals. All animals carry the *daf-2(e1370)* allele. The values for % egg laying post-dauer animals are presented as means. Each assay was repeated three times with 50 animals in each trial. n = 50.

deletion of the *mir-1* and *mir-44* seed sequences in the *tbc-7* 3'UTR, the recovered post-dauer adults exhibited a marked sterility compared to identical animals with an unmodified wild-type 3'UTR (Fig 4D). This underscores the importance of these miRNA binding sites and the need to block TBC-7 activity during the dauer stage to preserve germline integrity (Fig 4D). Furthermore, to complement these findings, we reasoned that the enhanced neuronal expression of *mir-1* or *mir-44* in *daf-2; aak(0)* mutants should suppress the AMPK germline defects by reducing the levels of *tbc-7*. Consistent with this, increased *mir-1* or *mir-44* expression in *daf-2; aak(0)* mutants suppressed the post-dauer sterility of AMPK mutants (Fig 4E). Together these results suggest that *mir-1* and *mir-44* play an important role in regulating RAB-7 activity by directly binding to the *tbc-7* 3'UTR to control its expression during the dauer stage.

## AMPK-mediated phosphorylation of TBC-7 could reduce its RabGAP activity

Although we have described how microRNAs contribute to the regulation of *tbc-7* in the neurons in response to AMPK activation, the kinetics of the *mir-1* and *mir-44* inhibition are not consistent with the timing of the first changes that occur in the germ line following the initiation of dauer formation. *mir-1* and *mir-44* mutants only exhibit post-dauer sterility after remaining in dauer for a longer duration, suggesting that these microRNAs may not be the sole regulators of TBC-7 activity. We therefore questioned whether AMPK might play an additional role in the regulation of TBC-7, perhaps through a key post-translational modification that would act more immediately than the activation of the miRNA-mediated *tbc-7* inhibition. We noticed that TBC-7 possesses a high stringency AMPK phosphorylation motif on Ser115 [35], making it a prime target of AMPK upon activation of the kinase during dauer formation to negatively affect the function or stability of TBC-7.

Very often, AMPK phosphorylates its targets thereby generating a 14-3-3 recognition site that eventually leads to target degradation [35]. To confirm if this is how AMPK affects TBC-7, we first performed a Western analysis on TBC-7 in *daf-2* control and *daf-2; aak(0)* mutant dauer larvae to determine if AMPK could affect the stability of TBC-7 by targeting it for subsequent degradation. If this is the case, the abundance of TBC-7 should be significantly higher in mutant animals that lack all AMPK signalling. Our analysis indicated that this was not the case and the abundance of TBC-7 was not significantly different between the two genotypes (Fig 5A). We then examined if the S520F mutation in *rr166* affected the stability of TBC-7. If the S520F mutation destabilized TBC-7, then we would expect a decrease in the levels of TBC-7 S520F compared to wild-type TBC-7. Western analysis showed that the S520F mutation did not affect the abundance of the TBC-7, suggesting that this mutation does not cause TBC-7 to be degraded at the onset of dauer in the absence of AMPK, but could disrupt its catalytic activity (Fig 5A'). These data indicate that neither AMPK nor the S520F mutation affects TBC-7 activity through targeted protein degradation, but rather that it must engage an alternative means of regulating TBC-7 activity prior to the onset of *mir-1* and *mir-44* expression/regulation (Fig 5A–5A'). However, since the abundance of TBC-7 was measured solely in the

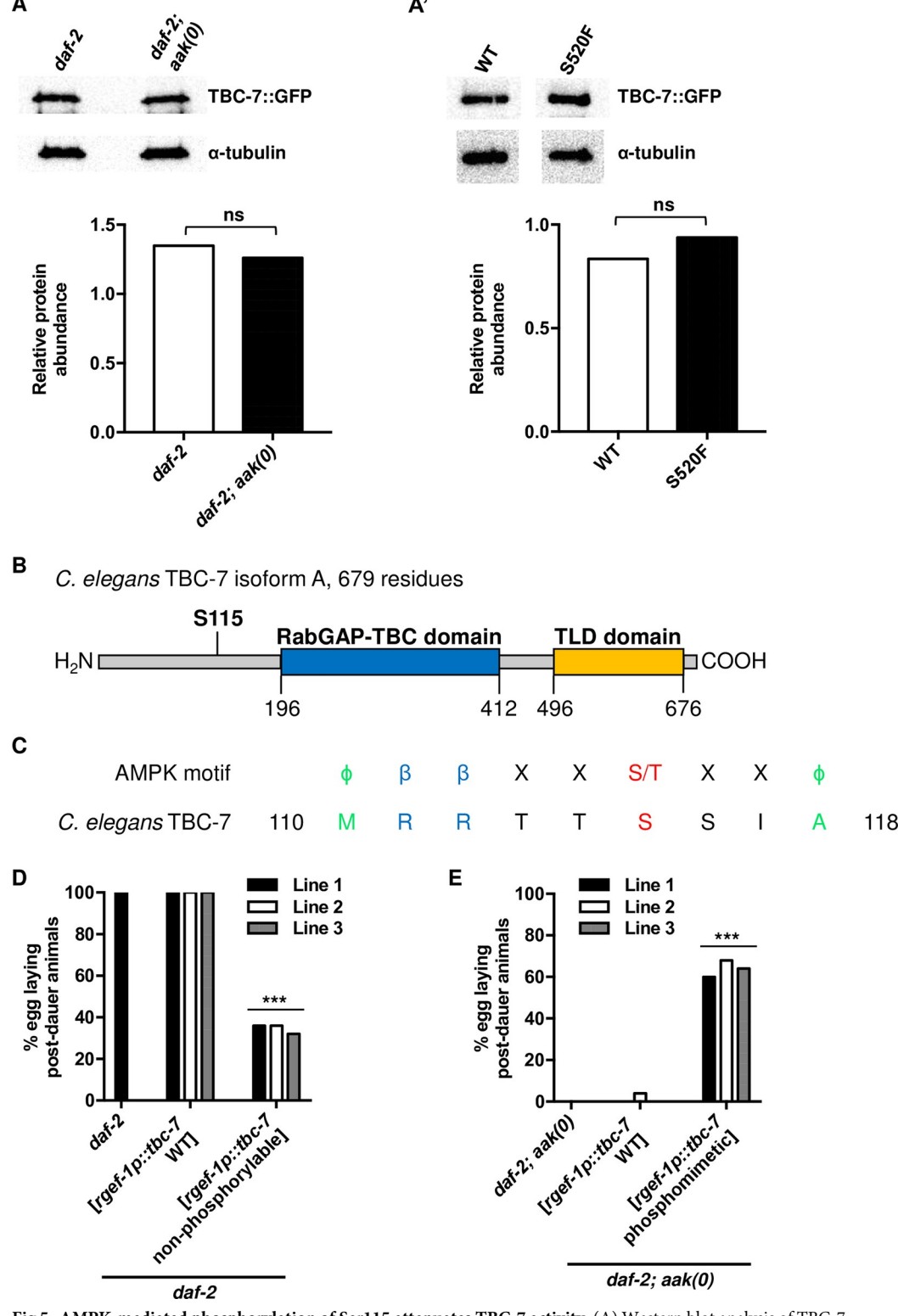

**Fig 5. AMPK-mediated phosphorylation of Ser115 attenuates TBC-7 activity.** (A) Western blot analysis of TBC-7 abundance shows no significant difference in the protein levels between the *daf-2* control and *daf-2; aak(0)* mutant backgrounds. TBC-7 expression was quantified and normalized to α-tubulin using ImageJ software. ns when compared to *daf-2* using Student's t-test. (A') Western blot analysis shows no significant difference in the protein levels between wild-type and S520F TBC-7. TBC-7 expression was quantified and normalized to α-tubulin using ImageJ software. ns when

compared to *daf-2* using Student's t-test. (B) A model showing the predicted domains and the consensus AMPK phosphorylation site of TBC-7. The RabGAP-TBC domain and TLD domain are shown in blue and yellow, respectively. The predicted AMPK phosphorylation site at Ser115 is indicated in bold [36]. (C) The amino acid sequence of TBC-7 was analysed to identify any potential AMPK phosphorylation motifs. The alignment of the sequence around Ser115 is shown with analogous sites to the AMPK phosphorylation motif sequence. Hydrophobic residues, green; hydrophilic residues, blue; phosphosite acceptor, red. (D) Animals expressing a S115A non-phosphorylable mutation of TBC-7 exhibit post-dauer sterility following recovery. ***P < 0.0001 when compared to *daf-2* animals and *daf-2* animals that express a wild-type version of *tbc-7* using Marascuilo procedure for % egg laying post-dauer animals. (E) A S115E phosphomimetic mutation of TBC-7 in *daf-2; aak(0)* mutants suppresses the post-dauer sterility. ***P < 0.0001 when compared to *daf-2* and *daf-2; aak(0)* mutants that express a wild-type variant of *tbc-7* using Marascuilo procedure for % egg laying post-dauer animals. All animals carry the *daf-2(e1370)* allele. The values for % egg laying post-dauer animals are presented as means. Each assay was repeated three times with 50 animals in each trial. n = 50.

neurons, we cannot exclude that the levels of TBC-7 may be affected in other tissues in animals that lack AMPK signalling.

AMPK has previously been shown to phosphorylate RabGAPs directly thereby inducing an intramolecular autoinhibition [37]. In mammalian cells, the AMPK-mediated phosphorylation of Ser168 near the N-terminus of TBC1D17 reduces its GAP activity toward its substrate Rab5 [37]. Due to the structural similarities between TBC1D17 and TBC-7, and the location of the AMPK phosphoacceptor sites, we questioned if the AMPK-mediated phosphorylation of TBC-7, like that of TBC1D17, could also reduce its GAP activity towards its target RAB-7. To examine the role of the predicted AMPK phosphorylation site Ser115 in the potential autoinhibition in TBC-7 (Fig 5B and 5C), we replaced the phosphoacceptor Ser115 with a non-phosphorylable alanine residue. Mutants expressing a non-phosphorylable variant of TBC-7 exhibited highly penetrant post-dauer sterility, consistent with this site being an important target of AMPK during the onset of the dauer stage (Fig 5D). To complement these findings, we reasoned that expressing the reciprocal phosphomimetic mutation of TBC-7 in *daf-2; aak(0)* mutants might compensate for the loss of AMPK in mutant animals, and suppress the mutant germline defects typically observed in these animals. Consistent with this possibility, the introduction of a variant TBC-7 where Ser115 is replaced with a negatively charged phosphomimetic glutamic acid residue was sufficient to suppress the germline defects in AMPK mutants (Fig 5E). Together, our data indicate that the AMPK-dependent phosphorylation of Ser115 on TBC-7 is important to antagonize its GAP activity, most likely toward RAB-7, thereby allowing RAB-7 to instruct the germ cells to execute quiescence and to ensure post-dauer reproductive competence.

## Discussion

Upon entry into the developmentally arrested dauer stage, AMPK modulates the germline cell cycle while also altering the chromatin landscape of these cells to instruct the appropriate changes in germline gene expression required to preserve germ cell integrity throughout the diapause [10]. In the absence of this master metabolic regulator, animals exhibit excessive germ cell hyperplasia during the dauer stage, while animals that are allowed to recover display highly penetrant sterility as well as various somatic defects [10]. This correlates with an increased abundance and abnormal distribution of both activating and repressive chromatin marks in the germ cells of these mutants. The modifications observed are likely responsible for the dramatic deviation from the normal expression of germline-expressed genes in AMPK mutant dauer larvae and this maladaptive expression program persists into the post-dauer animals [10].

Using genetic analysis to identify suppressors of this reproductive defect, we isolated 8 mutants that corresponded to 5 complementation groups. Two of these groups had multiple

alleles suggesting that the screen was near saturation, while two of these mutations corresponded to a gene encoding a RabGAP protein called TBC-7. *tbc-7* null mutants are non-viable, underscoring once again the power of unbiased forward genetic approaches in the identification of highly specific point mutations.

The chromatin marks that are abnormally regulated in the germ line of AMPK mutant animals are corrected in the *rr166* background. Curiously, the *rr289* allele that we isolated also shows germline hyperplasia in the dauer larva when compared to *aak(0)* mutants, but it still shows 61% fertility. Although we have not yet characterised this gene product, this allele is good evidence that the loss of germ cell integrity may not be directly linked to the germline hyperplasia. The germline phenotypes and the reduced dauer survival typical of AMPK mutants also appear unlinked. Although it has been hypothesized that the germ line hyperplasia may exhaust valuable lipid stores required for dauer survival, evidence suggests that this is unlikely. First, neither of the isolated *tbc-7* alleles suppress the premature dauer lethality that occurs in AMPK mutants. The timing of the two phenotypes is also very different; the hyperplasia happens during dauer formation, while the premature lethality occurs over a week later. Moreover, the reduced dauer survival of AMPK mutant dauer larvae was unaffected by restoring AMPK in the neurons [17]. Our data therefore suggest that the tissue-specific requirements for regulating germ cell integrity differ from those necessary for dauer survival, and that the neuronal activity of *tbc-7/rab-7* probably has no significant role in regulating triacylglyceride metabolism during the dauer stage.

*tbc-7* activity must be regulated to ensure the proper maintenance of germ cells during this period of quiescence. When *tbc-7* activity is compromised in AMPK defective animals, the defects in the germline chromatin landscape are corrected to near wild-type levels. This pathway may therefore be one of the major effectors of germline quiescence and integrity that occurs during the dauer stage. Curiously, the orthologues of this gene product are important for vesicle formation in the neurons in *Drosophila* and in mammals. Consistent with this, TBC-7 is expressed almost exclusively in the neurons of *C. elegans* and not in the germ cells.

The nervous system constitutes the perfect early response system to mediate any organismal adaptation to environmental challenges. Since AMPK activity in these cells is sufficient to change gene expression in the germ line, this protein kinase likely acts as a direct sensor for cellular ATP levels to transduce a signal from the neurons to vulnerable cell types dispersed throughout the organism [10,38].

*tbc-7* is expressed in many neurons in both dauer larvae and adult animals where it localizes to the late endosomal membrane to regulate the activity of RAB-7, a small *ras*-like GTPase involved in the positive regulation of the early to late endosome transition [26,39]. Late endosomes are involved in recycling cargo back to the plasma membrane, trafficking towards the Golgi body, or fusion with lysosomes to create a lysosome/late endosome hybrid to degrade cargo [26,39,40]. Given this wide range of functions, the mechanistic details of how RAB-7 may communicate information between the neurons and the germ line remain unclear. Our observations support a role for RAB-7 in the cellular trafficking of a neuron-derived diffusible signal to the germ line to establish germ cell quiescence during the dauer stage, although the nature of this diffusible signal remains enigmatic.

We have recently shown that small RNAs are involved in maintaining germ cell integrity during the dauer stage [10]. Upon the inhibition of endogenous siRNA activity, both the post-dauer fertility and the upregulation of chromatin marks in AMPK mutants are partially corrected. Furthermore, the loss of the dsRNA channel *sid-1* partially restores post-dauer germ cell integrity in the AMPK mutants, implying that the transfer of endogenous siRNAs during the dauer stage is misregulated and maladaptive in the absence of AMPK.

Our analysis suggests that this role of small RNAs may not be the sole required functions of these critical regulators during the various changes in gene expression that are associated with dauer formation. By analysing the microRNA seed sequences present within the 3'UTR of *tbc-7*, we were able to demonstrate how two microRNAs, *mir-1* and *mir-44*, are critical for the maintenance of germ cell integrity. Animals that lack *mir-1* or *mir-44* show the same germline defects as AMPK mutants, although these phenotypes are only visible after these mutants spend a longer period of time in the dauer stage. Moreover, these germline defects are further exacerbated upon the compromise of *rab-7* expression, suggesting that in the *mir-1* and *mir-44* mutants, *tbc-7* continually suppresses the activation of RAB-7. In other organisms, *mir-1* acts in a similar manner to regulate synaptic function at neuromuscular junctions [32]. These genetic data indicate that *mir-1* and *mir-44* are upstream negative regulators of *tbc-7* expression, thus they act indirectly as positive regulators of *rab-7* activity. Therefore, at the onset of dauer formation, AMPK activation might ultimately affect the formation of GTP-bound RAB-7, which in turn would be the major effector in the neurons by enabling communication with the germ cells to preserve their integrity over the long term.

It is however still unknown how or whether AMPK is able to regulate small RNA production or function, nor do we understand how this may bring about the observed changes in the chromatin modifications in the germ line. Interestingly, many components of the microRNA biogenesis machinery, including DRSH-1, PASH-1, and DCR-1, have multiple medium stringency AMPK phosphorylation motifs, which may suggest a role for AMPK in the direct regulation of the microRNA biogenesis machinery to ensure that microRNAs are produced in a timely manner. Alternatively, it is possible that microRNA production is regulated independently of AMPK/LKB1 signalling and these protein kinases impinge on components downstream of miRNA synthesis and processing to mediate these changes. Our initial identification of small RNA pathway involvement did not include regulators of the miRNA pathway, but this was due to their critical role in various aspects of embryonic and post-embryonic development [41]. A conservative explanation for these findings is that AMPK may be acting in a parallel pathway with *mir-1* and *mir-44* to restrict the activity of TBC-7, meanwhile the endogenous siRNA pathway or other small RNA populations might act further downstream to regulate the germline chromatin landscape and gene expression during the dauer stage. Further work will be required to discern if there are direct interactions between the miRNA pathway and AMPK signalling, although because of the critical role of miRNAs in developmental timing, and hence dauer formation, these pathways will have to be teased out carefully with newly available genetic tools in order to accurately interpret their potential involvement in these processes.

As larvae enter the dauer stage, the frequency of germ cell division decreases dramatically to finally arrest during the diapause. The observed kinetics of *mir-1* inhibition are insufficient to account for the timing of the changes in the germ line following the initiation of dauer formation. By using non-phosphorylable and phosphomimetic transgenic TBC-7 variants we showed that the AMPK-mediated phosphorylation of TBC-7 likely attenuates its GAP activity without altering its stability/abundance. Through analogy with other RabGAP AMPK targets, the AMPK-mediated phosphorylation may induce a conformational change in TBC-7, resulting in its autoinhibition [37]. Consistent with this, the removal of the AMPK phosphoacceptor site on TBC-7 results in a loss of germ cell integrity after only 2 days in the dauer stage, compared to the 7 days required for the same change to occur in *mir-1* and *mir-44* mutants. Therefore, AMPK likely acts as a direct inhibitor of TBC-7 activity, but it also acts successively by activating *mir-1* and *mir-44*, thereby enabling prolonged RAB-7 activity throughout the entire dauer stage; from onset to recovery. However, further experiments will need to be performed to confirm the direct phosphorylation of TBC-7 by AMPK. The AMPK-mediated phosphorylation of TBC-7 would account for the acute means of activating RAB-7 as animals begin to

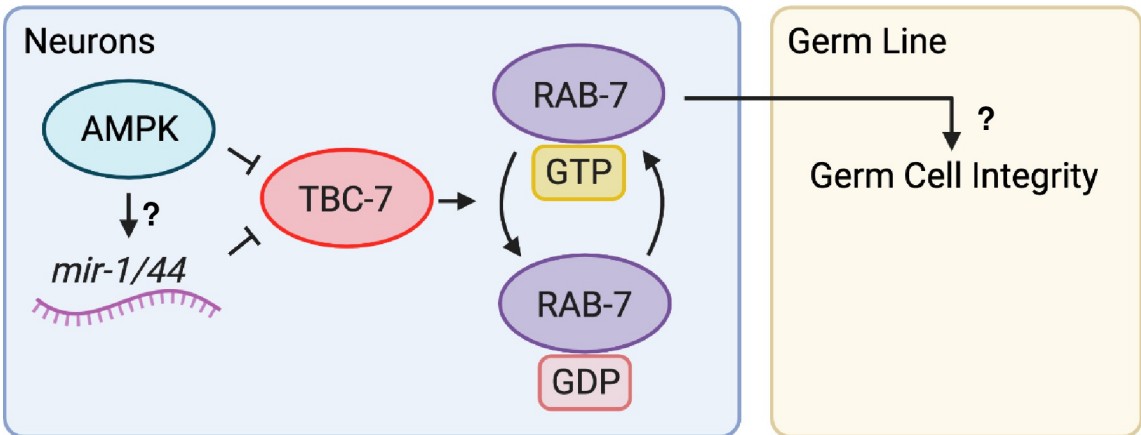

**Fig 6. AMPK regulates *rab-7* activity in neurons to establish germline quiescence and preserve germ cell integrity during the dauer stage.** When animals enter the dauer stage, AMPK, *mir-1*, and *mir-44* negatively regulate TBC-7 activity in the neurons. *mir-1* and *mir-44* are able to bind to the *tbc-7* 3'UTR to negatively regulate its expression. AMPK-mediated phosphorylation of TBC-7 negatively regulates its activity. In parallel, it may be possible that AMPK could act directly on the miRNA biogenesis machinery to promote the production of *mir-1* and *mir-44*, as suggested by the presence of phosphorylation motifs on microRNA biogenesis components. It is still unknown which particular neuronal *rab-7* pathway alters the chromatin landscape in the germ cells of the dauer larva downstream of this mechanism. Reducing TBC-7 activity allows RAB-7 to remain in its active GTP-bound form to mediate changes in the dauer germ line to preserve germ cell integrity and post-dauer reproductive fitness.

form dauer larvae, while the *mir-1* and *mir-44* regulation of the *tbc-7* transcript would act redundantly and later during the diapause to ensure that any remaining mRNA cannot be expressed to antagonize the critical RAB-7 activation necessary for germline integrity during the later stages of the diapause.

In summary, we have characterised an AMPK-dependent microRNA regulated *tbc-7/rab-7* pathway that acts cell non-autonomously in the neurons to establish quiescence in the germ line during periods of energetic stress. As the animals enter the dauer stage, AMPK phosphorylates Ser115 on TBC-7 to affect its GAP activity, thus allowing RAB-7 to remain in its GTP-bound active form and to perform its function, while later in the diapause, *mir-1* and *mir-44* bind to the 3'UTR of *tbc-7* to block its expression (Fig 6).

Although we have identified this *mir-1/mir-44/tbc-7/rab-7* pathway, we have yet to uncover which particular function of *rab-7* is responsible for transmitting this dauer signal from the neurons to the germ line. Nor do we understand how a *rab-7* pathway occurring in the neurons can alter the chromatin landscape in the germ cells of the dauer larva. Communication between the soma and the germ cells is a conserved phenomenon [42]. However, it remains undetermined whether this transfer of information is mediated through a RAB-7-dependent neuroendocrine pathway or via an alternative means of intertissue communication that has yet to be fully characterized.

## Materials and methods

### *C. elegans* strains and maintenance

All *C. elegans* strains were maintained at 15˚C on nematode growth medium petri plates and according to standard protocols unless otherwise indicated [43]. The strains used in this study are listed in S6 Table.

Transgenic lines and compound mutants were created in the laboratory using standard molecular genetics approaches. In this manuscript, we list *daf-2(e1370) aak-1(tm1944) III; aak-2(ok523) X* mutants as *daf-2; aak(0)* mutants to indicate a complete lack of AMPK

catalytic activity. Since *aak(0)* is 'composed' of two genes on two different chromosomes, we keep it on a 'separate' chromosome when writing the genotype, detached from *daf-2(e1370)* on chromosome III and *tbc-7(rr166)* on chromosome X. All mutant strains isolated following mutagenesis were backcrossed at least 5 times prior to characterization and subsequent whole-genome sequencing. All transgenes were injected at 15 ng/μL unless specified otherwise. The transgene containing *pre-mir-1* driven by *rgef-1* promoter were injected at 1 ng/μL. The transgene containing *pre-mir-44* driven by *rgef-1* promoter was injected at 0.5 ng/μL. All injection mixes contained 120 ng/μL of pRF-4 (*rol-6D*) as a dominant injection marker and the concentration of DNA was made up to 200 ng/μL with an empty vector pSK.

## Genetic suppressor screen

Animals were mutagenized as described elsewhere with some modifications [43]. $P_0$ L4 AMPK null (*daf-2; aak(0);* strain name MR1000) mutants were treated with 0.05 mM EMS. $F_1$ animals were isolated to separate the $F_1$ generation from the $P_0$ generation. The $F_1$ generation was allowed to grow to become gravid adults before they were synchronized to obtain $F_2$ candidates that were homozygous for the EMS-generated mutation. The $F_2$ generation was assessed for their ability to suppress the AMPK germline defects. $F_2$ animals were switched to a restrictive 25°C temperature for 48 hours to induce dauer formation and kept for an additional 48 hours in the dauer stage for a total of 96 hours. All $F_2$ animals were subjected to 1% SDS in order to eliminate dauer-defective mutants. After, $F_2$ animals were switched into the permissive temperature of 15°C to trigger dauer recovery. The $F_2$ generation was screened for their ability to suppress germline defects typical of AMPK mutants; candidates were assessed on their % of egg-laying post-dauer animals, No. of germ cells in dauer larvae, post-dauer brood size, and % of adult animals with wild-type appearance (the lack of any vulval defects or premature death). In total, 8 independent alleles were isolated from approximately 7000 haploid mutagenized genomes.

Potential candidates that were able to suppress the AMPK germline defects were submitted for next-generation sequencing using Deep Sequencing Data Shotgun (Beads 360). Genomic DNA was extracted using the phenol-chloroform DNA isolation method. The integrity of the extracted DNA was examined on an agarose gel and through nanodrop spectrophotometer. Library preparation and quality check was done by the Génome Québec sequencing facility. Seven candidates were submitted for sequencing along with the starting mutagenesis strain *daf-2; aak(0)* (strain name: MR1000). The average number of reads per candidate is 58,268,582 reads. The sequencing results of each candidate and the starting strain *daf-2; aak(0)* were aligned to WBcel235 as a reference genome. After assembly of the genomes, all the candidates were compared against *daf-2; aak(0)* to identify any potential mutations caused by the EMS. Variant calling was done using bcftools on each candidate after genome alignment. Each genetic variation was ranked using the Z-score from Wilcoxon rank sum test of mutant alleles vs. the reference genome sequence.

## Complementation and other genetic analyses

Complementation analysis was conducted as previously described [44]. Each candidate was injected with a plasmid containing GFP driven by the pharyngeal promoter *myo-2p* and heat shocked at 30°C for 6 hours to induce males. Males expressing pharyngeal GFP were crossed with an EMS-generated mutant adult hermaphrodite (that did not express pharyngeal GFP) of a different genotype. Cross progeny was identified using the pharyngeal GFP marker. The $F_1$ cross progeny that were heterozygous for two EMS-induced alleles were made to transit through the dauer stage and the % of fertile post-dauer animals and the No. of germ cells in

dauer larvae were assessed as a measure of complementation. All eight candidates generated from the EMS screen were crossed to each other in order to assign their complementation group. EMS-generated candidates that generated cross progeny that were post-dauer sterile or suffered from germline hyperplasia were categorized into different complementation groups, while those that were post-dauer fertile and could suppress their germline hyperplasia were categorized into the same complementation group.

To determine if *rr166* is a null allele of *tbc-7*, we crossed *rr166* into a known *tbc-7(tm10766)* null background. The *tbc-7(tm10766)* null mutation is embryonic lethal, so these mutants are balanced and maintained as heterozygotes. When *tm10766* hermaphrodites were crossed with *tbc-7(rr166)* males, we were able to isolate viable F1 mutants that carried the *tbc-7(tm10766)* deletion and the *tbc-7(rr166)* point mutation, suggesting that *rr166* is a hypomorphic allele of *tbc-7* (S4 Fig).

### Dauer recovery assay

Post-dauer fertility and somatic phenotypes were assayed and quantified as described elsewhere [10]. A population of genetically identical mutants were synchronized using alkaline hypochlorite. The resulting embryos were hatched in M9 buffer and then plated on NGM plates seeded with OP50. The resulting animals were incubated at 25˚C for 96 hours so that the animals can spend at least 24 hours in the dauer stage. Afterwards, the animals were switched back to a permissive temperature of 15˚C to trigger dauer recovery and allow for normal development. The animals were then singled onto NGM plates. The post-dauer fertility and brood size of each animal were assessed 7 days after switching into the permissive temperature. The brood size of each animal is the number of hatched progenies. The post-dauer adults were assessed visually for any post-dauer somatic defects.

### DAPI staining and germ cell quantification

DAPI staining and germ cell quantification of the dauer larvae was conducted as described elsewhere [10]. A population of genetically identical animals were synchronized and allowed to hatch in M9 solution. The animals were then plated onto NGM seeded with OP50 and incubated at 25˚C for 96 hours to allow the animals to spend at least 24 hours in the dauer stage. The resulting dauer larvae were washed off the plate and soaked in Carnoy's solution (60% ethanol, 30% acetic acid, 10% chloroform) overnight on a shaker. Afterwards, the Carnoy's solution was removed and the dauer larvae was washed twice with 1x PBS + 0.1% Tween 20 (PBST). The dauer larvae were then stained with 0.1 mg/mL DAPI solution for 30 minutes on a shaker. Finally, the DAPI solution was removed, and the sample was washed four times with PBST (15 minutes for each wash on a shaker). The larvae were then mounted onto glass slides. Germ cells per dauer gonad was determined based on their position and their nuclear morphology. The number of germ cells in the dauer larvae were counted manually.

### Dauer survival assay

Dauer survival assay was conducted as described elsewhere [8]. A population of genetically identical animals were synchronized and allowed to hatch in M9 solution. The animals were then plated onto NGM seeded with OP50 and incubated at 25˚C for 48 hours to induce dauer formation. Then dauer larvae were singled into PCR strip tubes containing 10 µL of M9 buffer, one animal per well. Trapping the larvae in PCR tubes prevented the dauer larvae from crawling off the bacterial lawn and desiccating at the edge of the plate. Dauer survival was monitored daily and was scored based on their movement in response to exposure to a focused beam of 425–440 nm light.

## Western blot analysis

Western blot analysis was conducted as described elsewhere [10]. *C. elegans* dauer larvae and adults were manually picked into 10 μL PBST. Then 10 μL of SDS loading buffer (5% β-mercaptoethanol, 0.02% bromophenol blue, 30% glycerol, 10% sodium dodecyl sulfate, 250 mM pH 6.8 Tris-Cl) was added before the entire mixture was subjected to multiple rounds of freeze-thawing with liquid nitrogen and 100°C heat block. Nitrocellulose membranes were incubated with rabbit anti-GFP (homemade antibody), rabbit anti-H3K4me3 (1:1000 dilution, Abcam, ab8580), rabbit anti-H3K9me3 (1:1000 dilution, Cell Signalling Technology, 9754S), or mouse anti-α-tubulin (1:2000, Sigma-Aldrich, St. Louis, MO, USA). After SDS-PAGE and Western blotting, proteins were visualized using horseradish-peroxidase-conjugated anti-rabbit or anti-mouse secondary antibodies (1:2000, Bio-Rad, Hercules, CA, USA).

## Fosmid preparation and injection

The fosmid containing a wild-type copy of *tbc-7* translationally fused to GFP is from the UBC N2 fosmid library (Don Moreman) that was ordered from the TransgeneOme Project *C. elegans*. The clone identification is: 3304493055384826 E02. This fosmid contains the entire sequence of the *tbc-7* gene, including exons and introns and the 5' and 3'UTRs. It also contains an additional upstream sequence (starts X: 5,120,813) that is 13,467 bp in size starting from the transcription start site and an additional downstream sequence (ends X: 5,154,839) that is 8,095 bp in size starting from the end of the last exon. The fosmid was extracted from *E. coli* using the EZ-10 Spin Column Plasmid DNA Miniprep Kit (Cat. #: BS513) from Bio Basic. The purified fosmid was co-injected with a transformation marker pRF-4 (plasmid containing a dominant negative variant of the *rol-6* gene). The fosmid was injected at a concentration of 1 ng/μL and the pRF-4 transformation marker was injected at a concentration of 120 ng/μL. The injections were performed on *daf-2* mutants for the imaging of TBC-7::GFP in Fig 2B and 2C, and on *daf-2; aak(0); tbc-7* mutants to revert the suppression conferred by the *rr166* allele in Fig 2A.

## Plasmid cloning and preparation

To generate the plasmid that drives a GTP-locked variant of RAB-7 in the neurons, the GTP-locked variant of RAB-7 Q68L was amplified from a pGEX-5X-2 vector containing RAB-7 Q68L (a generous gift from Dr. Christian Rocheleau). The sequences of the primers used to amplify *rab-7* are 5'-atggatgaactatacaaaatgtcgggaaccagaaagaa-3' and 3'-acacggttaacaattgcatccc-gaattctgctggttctg-5'. The GTP-locked variant of RAB-7 Q68L was inserted between the neuronal promoter *rgef-1* and a wild-type *rab-7* 3'UTR in the pSK vector using Gibson assembly.

The 3'UTR deletions were generated by amplifying a wild-type copy of the *tbc-7* 3'UTR from genomic DNA and inserting it into an empty vector pMR377. The sequences of the primers used to amplify the 3'UTR are 5'-tcgtagaattccaactgagcgcattcactctgcccaag-3' and 3'-ccgtacggccgactagtaggttcaggctgcaagaaaaaca-5'. The *mir-1* and *mir-44* binding sites on the *tbc-7* 3'UTR were removed by linearizing the entire pMR377+3'UTR plasmid using a set of primers that flanked the *mir-1* and *mir-44* seed sequences in a PCR reaction. The linearized PCR product was phosphorylated by polynucleotide kinase for 30 minutes and ligated by DNA ligase overnight to generate a plasmid that contains the *tbc-7* 3'UTR with the *mir-1* and *mir-44* seed sequences removed. The sequences of primers used to delete the seed sequences are 5'-ctcctcgcctccaatgtttg-3' and 3'-ggcaatacttaagaatgaggaagg-5'. The deletion from the site directed mutagenesis was verified by sequencing the pMR377 plasmid with the mutant *tbc-7* 3'UTR. The mutant *tbc-7* 3'UTR was amplified using the original set of primers for the 3'UTR and

inserted using Gibson assembly behind a wild-type copy of *tbc-7* cDNA driven by the *rgef-1p* neuron-specific promoter in a modified pPD95.77 vector.

The amino acid substitutions to mutate the predicted AMPK phosphosite on TBC-7 was done by amplifying a wild-type copy of *tbc-7* from cDNA and inserting it into an empty vector pMR377. The sequences of the primers used to amplify *tbc-7* are 5'-ctttacattttgttttcagaatgacg-gaaaacgctggatc-3' and 3'-cagttggaattctacgaatgttagtcgctggaagtaacatgga-5'. Site directed mutagenesis was performed on the pMR377+*tbc-7* plasmid to replace the Ser115 residue with either Ala or Glu. Primers were designed around the AGT codon that encodes for Ser115. One primer contained overlaps with the GCC codon or CTC codon that encodes for Ala or Glu, respectively. The plasmid containing *tbc-7* was mutagenized using these primers to create a linearized PCR product. This product was phosphorylated by polynucleotide kinase for 30 minutes and ligated by DNA ligase overnight to generate a plasmid that contains a mutant variant of *tbc-7* that has its Ser115 replaced with either S115A or S115E. The primers used to create the S115A mutation are 5'-aattttgattgaaagaagtcggatc-3' and 3'-GGCgagcactccaccttcatctg-5'. The primers used to create the S115E mutation are 5'-aattttgattgaaagaagtcggatc-3' and 3'-CTCgagcactccaccttcatctg-5'. The mutant *tbc-7* gene was amplified using the original set of primers used to amplify the *tbc-7* cDNA sequence and was inserted using Gibson assembly between a *rgef-1p* neuron-specific promoter and wild-type *tbc-7* 3'UTR in a modified pPD95.77 vector.

All plasmids used in this manuscript were verified using restriction digestion analysis and sequencing analysis.

## Immunostaining and quantification

*C. elegans* dauer larvae and post-dauer adult gonads were dissected, fixed, and stained as described elsewhere [45]. Extruded gonads were incubated with rabbit anti-H3K4me3 (1:500 dilution, Abcam, ab8580) or rabbit anti-H3K9me3 (1:500 dilution, Cell Signalling Technology, 9754S). Secondary antibodies were Alexa-Fluor-488–coupled goat anti-rabbit (1:500; Life Technologies, Carlsbad, CA, USA). Gonads were counterstained with DAPI (0.1 μg/mL dilution, Roche Diagnostics, 10236276001). Microscopy was performed as described elsewhere [10,46]. Ratios for the fluorescence intensity across the germ line were determined using ImageJ. The ratio was calculated by dividing the immunofluorescence signal of either anti-H3K4me3 or anti-H3K9me3 by the fluorescence signal of DAPI per nucleus.

## RNA interference by feeding

RNA interference was conducted as described elsewhere based on standard feeding protocols [10,47]. Bacterial clones expressing dsRNA from the Ahringer *C. elegans* RNAi feeding library were grown in LB medium with ampicillin at 37°C overnight [48]. The bacterial culture was then seeded onto NGM plates containing ampicillin and IPTG (1 mM) and allowed to grow for 48 hours at room temperature to induce dsRNA expression. A population of genetically identical animals were synchronized and allowed to hatch in M9 buffer before they were plated onto NGM plates containing bacteria that expressed dsRNA. The animals were incubated at 25°C for 96 hours to allow the animals to spend at least 24 hours in the dauer stage. If the experiment required the animals to be separated, individual animals were separated onto NGM plates containing bacteria that expressed dsRNA.

## Generation of tissue-specific RNAi strains

All tissue-specific RNAi strains contain a *rde-1(mkc36)* mutation. *rde-1(mkc36)* contains a 67 bp insertion and a 4 bp deletion that creates three premature stop codons [24]. *rde-1(mkc36)*

mutants do not respond to exogenous RNAi by feeding or injection [24]. A germline-specific RNAi strain was generated by expressing a single-copy of *rde-1* in the germ line driven by the *sun-1* promoter inserted on chromosome II using MosSCI [24]. To generate neuron-specific RNAi strains, *rde-1* and *sid-1* were expressed exclusively in the neurons of *rde-1(mkc36)* animals using the *rgef-1* promoter [49]. Neurons of *C. elegans* exhibit weak penetrance through feeding, thus extra copies of neuronal *sid-1* potentially serves as a dsRNA sink, allowing for a robust RNAi phenotype through feeding [23]. Tissue-specific RNAi strains do not exhibit any developmental defects when grown in replete conditions, when passed through the dauer stage on standard NGM plates, or when grown on L4440 empty vector control compared to their non-*rde* counterpart. The specificity and efficiency of these tissue-specific RNAi strains were examined by feeding animals expressing either neuronal or germline TBC-7::GFP with either empty vector control L4440 or dsRNA against GFP. The levels of TBC-7::GFP expression were quantified using Western blotting against GFP (S8A and S8B Fig).

## Prediction of *mir-1* and *mir-44* seed sequence

*mir-1* and *mir-44* were identified as a potential regulator of *tbc-7* through TargetScanWorm release 6.2 (S5 Table). Two highly conserved *mir-1* seed sequences and one highly conserved *mir-44* seed sequence were identified in the 3'UTR of *tbc-7*. The $P_{CT}$ and conserved branch length are measures of the biological relevance of the predicted miRNA and target interaction, with greater values being more likely to have detectable biological function [50,51].

## Supporting information

**S1 Fig. Mutants partially suppress the dauer germline hyperplasia and post-dauer sterility associated with loss of AMPK.** (A-C) Mutants isolated from an EMS suppressor screen partially suppress the (A) post-dauer sterility, (B) dauer germline hyperplasia, and (C) brood size defects associated with a lack of AMPK signalling. \*\*\*P < 0.0001, \*\*P < 0.001, \*P < 0.05 when compared to *daf-2; aak(0)* using ordinary one-way ANOVA for post-dauer brood size and no. of germ cells in dauer larvae. \*\*\*P < 0.0001 when compared to *daf-2; aak(0)* using Marascuilo procedure for % egg laying post-dauer animals. All animals isolated from the screen include *daf-2; aak(0)* in the background. The values for % egg laying post-dauer animals, no. of germ cells in dauer larvae, and post-dauer brood size are presented as means. Each assay was repeated three times with 50 animals in each trial. n = 50.
(TIF)

**S2 Fig. The abnormal abundance and distribution of chromatin marks typical of dauer larvae that lack AMPK is corrected in *rr166* mutants.** (A) Global levels of H3K4me3 and H3K9me3 were quantified by performing whole-animal western analysis on dauer larvae. (A'-A") Levels of chromatin marks were quantified and normalized to α-tubulin using ImageJ software. \*\*\*P < 0.0001 when compared to *daf-2; aak(0)* using Student's t-test. (B-C") The distribution and abundance of activating and repressive chromatin marks are corrected in the *daf-2; aak(0); rr166* mutants. The top row (*daf-2*), middle row (*daf-2; aak(0)*), and bottom row (*daf-2; aak(0); rr166*) show (B, B', B") H3K4me3 (green), (C, C', C") H3K9me3 (green), and DAPI (red). The graphs represent the average immunofluorescence signal of anti-H3K4me3 and anti-H3K9me3 normalized to DAPI in each nucleus across the dissected germ line. All images are merged, condensed Z stacks and are aligned such that distal is left and proximal is right. Due to technical difficulties, only single gonadal arms were analysed (distal, proximal). \*\*P < 0.001 using the F-test for variance when compared to *daf-2; aak(0)*. All animals carry

the *daf-2(e1370)* allele. Scale bar: 4 μm. n = 15.
(TIF)

**S3 Fig. The *rr166* mutation restores appropriate chromatin remodeling in post-dauer AMPK mutants.** (A) Global levels of H3K4me3 and H3K9me3 were quantified by performing whole-animal western analysis of dauer larvae. (B) Chromatin marks were quantified and normalized to α-tubulin using ImageJ software. ***P < 0.0001 when compared to *daf-2; aak(0)* using Student's t-test. (B-C") The aberrant distribution and abundance of activating (H3K4me3) and repressive (H3K9me3) chromatin marks observed in post-dauer adults that lack AMPK signalling are corrected in the *daf-2; aak(0); rr166* mutants. The top row (*daf-2*), middle row (*daf-2; aak(0)*), and bottom row (*daf-2; aak(0); rr166*) show (B, B', B") H3K4me3 (green), (C, C', C") H3K9me3 (green), and DAPI (red). The graphs represent the average immunofluorescence signal of anti-H3K4me3 and anti-H3K9me3 normalized to DAPI across the dissected germ line. All images are merged, condensed Z stacks that are aligned such that distal is left and proximal is right. Due to technical difficulties, only single gonadal arms were analysed (distal, proximal). **P < 0.001 using the F-test for variance when compared to *daf-2; aak(0)*. All animals carry the *daf-2(e1370)* allele. Scale bar: 4 μm. n = 15.
(TIF)

**S4 Fig. *tbc-7(rr166)* is a hypomorphic allele.** (A) Schematic showing the cross with *tbc-7 (rr166)* with *tbc-7(tm10766)*, which contains a 21,846 bp deletion that completely removes *tbc-7*. Homozygous *tbc-7(tm10766)* is non-viable and is balanced with *tmC30*, which is a balancer that has a recessive Long (Lon) phenotype. To confirm if *tbc-7(rr166)* is a hypomorphic allele, *tbc-7(rr166)* was crossed with *tbc-7(tm10766)* and the $F_2$ cross progeny were examined in order to identify $F_1$ heterozygous *tbc-7(rr166)/tbc-7(tm10766)* hermaphrodites. $F_1$ heterozygous *tbc-7(rr166)/tbc-7(tm10766)* hermaphrodites should yield no Lon phenotype progeny, while $F_1$ heterozygous *tbc-7(rr166)/tmC30* should yield 25% Lon phenotype progeny. (B) Table showing the percentage of $F_1$ genotypes over two independent crosses. The $F_2$ progeny of every $F_1$ was assessed. $F_1$ genotypes assessed per cross n $\geq$ 150.
(TIF)

**S5 Fig. TBC-7 does not work in the body wall muscle or excretory system to regulate germ cell integrity.** A wild-type copy of *tbc-7* cDNA expressed exclusively in the (A) muscles by the *myo-3* promoter or the (B) excretory system by the *sulp-5* promoter in the *tbc-7*-suppressed mutants does not revert the suppression of the AMPK germline phenotypes. ns when compared to *daf-2; aak(0); tbc-7* based on Marascuilo procedure for % egg laying animals. All animals carry the *daf-2(e1370)* allele. The values for % egg laying post-dauer animals are presented as means. Each assay was repeated three times with 50 animals in each trial. n = 50.
(TIF)

**S6 Fig. *rab-7* functions in the neurons while *rab-10* functions in the germ line.** (A) Tissue-specific RNAi experiments in the *daf-2; aak(0); tbc-7* mutant show that *rab-7* functions in the neurons while *rab-10* functions in the germ line. ***P < 0.0001 when compared to L4440 empty vector using Marascuilo procedure for % of egg laying post-dauer animals. The values for % egg laying post-dauer animals are presented as means. Each assay was repeated three times with 50 animals in each trial. n = 50.
(TIF)

**S7 Fig. *mir-1; mir-44; daf-2* double mutants display the same post-dauer germline defects.** (A) *mir-1; mir-44; daf-2* mutants exhibit post-dauer sterility after 7 days in the dauer stage. ***P < 0.0001, **P < 0.001, *P < 0.01 when compared to *daf-2* animals that spent an identical

duration in the dauer stage using Marascuilo procedure for % egg laying animals. (B) Reducing *rab-7* levels greatly enhances/accelerates the post-dauer sterility associated with a loss of *mir-1* and *mir-44*. ***P < 0.0001 when compared to L4440 empty vector using Marascuilo procedure for % egg laying animals. All animals carry the *daf-2(e1370)* allele. The values for % egg laying post-dauer animals are presented as means. Each assay was repeated three times with 50 animals in each trial. n = 50.
(TIF)

**S8 Fig. Tissue-specific TBC-7::GFP used as a sensor for RNAi efficiency and specificity in the neurons and the germ line.** TBC-7::GFP was expressed in either the neurons (*rgef-1* promoter) or the in the germ line (*sun-1* promoter) of animals with (A) germline-specific RNAi and (B) neuron-specific RNAi. Tissue-specific RNAi animals were treated with either empty vector control L4440 or with dsRNA against *tbc-7*. The levels of TBC-7::GFP protein expression were quantified using Western blotting against GFP. α-tubulin was used as a loading control.
(TIF)

**S1 Table. Mutants isolated from an EMS suppressor screen partially suppress the post-dauer somatic defects of AMPK mutants.** Mutants were allowed to transit through the dauer stage and recover. The post-dauer somatic defects were assessed seven days after the recovery period. All animals carry the *daf-2(e1370)* allele. Mutants isolated from the EMS screen are *daf-2; aak(0)*. The values are presented as means. Each assay was repeated three times with 50 animals in each trial. n = 50.
(TIF)

**S2 Table. *daf-2; aak(0); rde-1* mutants with tissue-specific expression of *rde-1* exhibit tissue-specific RNAi phenotypes.** To create a tissue-specific RNAi strain in the *daf-2; aak(0); rde-1* background, *rde-1* was driven exclusively in the neurons using a *rgef-1* promoter or in the germ line using a *sun-1* promoter to create a neuron-specific or germline-specific RNAi strain, respectively. To confirm that *daf-2; aak(0); rde-1* mutants with tissue-specific expression of *rde-1* exhibit tissue-specific RNAi phenotypes, mutants were fed dsRNA against *dpy-10* (hypodermis), *egg-5* (germ line), or *unc-13* (neurons) and the phenotypes were scored. Each dsRNA treatment exhibits a unique phenotype, such as dumpy (*dpy-10* RNAi), embryonic lethal (*egg-5* RNAi), or paralysis (*unc-13* RNAi). *daf-2* control, *daf-2; aak(0)*, or the tissue-specific RNAi strains were synchronized and plated on bacteria expressing one of three dsRNAs. These animals transited through the dauer stage, and their RNAi phenotype was scored as post-dauer adults. Only mutants expressing *rde-1* in the same tissues targeted by the dsRNA should exhibit the RNAi phenotype. All mutants are *daf-2; aak(0)* except the *daf-2* control. The values are presented as means. Each assay was repeated three times with 50 animals in each trial. n = 50.
(TIF)

**S3 Table. Post-dauer fertility after RNAi treatment against *rab* genes.** *daf-2* control and *daf-2; aak(0); tbc-7* mutants were fed dsRNA against all known and predicted *rab* genes then allowed to transit through the dauer stage. L4440 empty vector was used as a control. All animals have the *daf-2(e1370)* allele. The values for % egg laying post-dauer animals are presented as means. Each assay was repeated three times with 50 animals in each trial. n = 50.
(TIF)

**S4 Table. *daf-2; aak(0); rde-1; tbc-7* mutants with tissue-specific expression of *rde-1* exhibit tissue-specific RNAi phenotypes.** To create a tissue-specific RNAi strain in the *daf-2; aak(0);*

*rde-1; tbc-7* background, *rde-1* was driven exclusively in the neurons using a *rgef-1* promoter or in the germ line using a *sun-1* promoter to create a neuron-specific or germline-specific RNAi strain, respectively. To confirm that *daf-2; aak(0); rde-1; tbc-7* mutants with tissue-specific expression of *rde-1* exhibit tissue-specific RNAi phenotypes, mutants were fed dsRNA against *dpy-10* (hypodermis), *egg-5* (germ line), or *unc-13* (neurons) and the phenotypes were scored. Each dsRNA treatment exhibits a unique phenotype, such as dumpy (*dpy-10* RNAi), embryonic lethal (*egg-5* RNAi), or paralysis (*unc-13* RNAi). *daf-2* control, *daf-2; aak(0)*, or the tissue-specific RNAi strains were synchronized and plated on bacteria expressing one of three dsRNAs. These animals transited through the dauer stage, and their RNAi phenotype was scored as post-dauer adults. Only mutants expressing *rde-1* in the same tissues targeted by the dsRNA should exhibit the RNAi phenotype. All mutants are *daf-2; aak(0); tbc-7* except the *daf-2* control. The values are presented as means. Each assay was repeated three times with 50 animals in each trial. n = 50.
(TIF)

**S5 Table. Predicted *mir-1* and *mir-44* seed sequences in the 3' UTR of *tbc-7*.** Two highly conserved *mir-1* seed sequences (bold) and one highly conserved *mir-44* seed sequence (bold) were identified in the 3' UTR of *tbc-7* (TargetScanWorm release 6.2), suggesting that *mir-1* and *mir-44* directly regulates *tbc-7* expression. The $P_{CT}$ and conserved branch length are measures of the biological relevance of the predicted miRNA and target interaction, with greater values being more likely to have detectable biological function [50,51].
(TIF)

**S6 Table. A list of strains used in this study.**
(DOCX)

**S1 Data File. Data availability document.**
(XLSX)

## Acknowledgments

We thank the members of the Roy and Zetka laboratories for their thoughtful discussions and comments on the manuscript. We acknowledge Anja Boskovic for help with the bioinformatics and Julian Moran for technical assistance during the screen. We also acknowledge the *Caenorhabditis elegans* Genetics Center and the National BioResource Project: *Caenorhabditis elegans* Shigen for *C. elegans* strains.

## Author Contributions

**Conceptualization:** Christopher Wong, Pratik Kadekar, Elena Jurczak, Richard Roy.

**Data curation:** Christopher Wong, Pratik Kadekar, Richard Roy.

**Formal analysis:** Christopher Wong, Pratik Kadekar, Richard Roy.

**Funding acquisition:** Richard Roy.

**Investigation:** Christopher Wong, Pratik Kadekar, Elena Jurczak, Richard Roy.

**Methodology:** Christopher Wong, Richard Roy.

**Supervision:** Richard Roy.

**Validation:** Christopher Wong.

**Writing – original draft:** Christopher Wong, Richard Roy.

**Writing – review & editing:** Christopher Wong, Richard Roy.

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
