## [Decision Letter · Decision Letter 0]

4 Jan 2023

Dear Dr Roy,

Thank you very much for submitting your Research Article entitled 'Germline stem cell integrity and quiescence are controlled by an AMPK-dependent neuronal trafficking pathway' to PLOS Genetics.

The manuscript was seen by four peer reviewers, one of whom previously evaluated it for Review Commons. (All reviewers were aware of and had access to the manuscript history at Review Commons). As you will see, the reviewers are generally positive, but raise a number of concerns having to with interpretation and presentation.

The manuscript and the reviews have now been considered by members of the editorial board. Overall, we agree with the reviewers' comments, and invite submission of a revised manuscript. We do not think any new experiments are needed but we ask that you address all of the reviewers' concerns with changes to figures, text, interpretation, and organization. In particular, for a revised manuscript to move forward, it will be necessary to be critical and rigorous with regard to interpretation, add a great deal of additional detail to the methods, properly and completely label figures and axes, and be precise and comprehensive with regard to genotypes when describing results.

We therefore ask you to modify the manuscript according to the review recommendations. Your revisions should address the specific points made by each reviewer.

We hope to receive your revised manuscript within the next 60 days. If you anticipate any delay in its return, we would ask you to let us know the expected resubmission date by email to plosgenetics@plos.org.

Yours sincerely,

Gregory S. Barsh

Editor-in-Chief

PLOS Genetics

Gregory Copenhaver

Editor-in-Chief

PLOS Genetics

Reviewer's Responses to Questions

**Comments to the Authors:**

Reviewer #1: In the manuscript entitled “Germline stem cell integrity and quiescence are controlled by an AMPK-dependent neuronal trafficking pathway”, Wong et al. present the next chapter in our understanding of how AMPK coordinates dauer diapause programs in C. elegans. During early larval stages, unfavorable conditions (or reduced insulin signaling in the daf-2 mutant) drive animals to enter the dauer diapause stage. During diapause, dauer larvae remodel their bodies to become stress resistant and pause their metabolic and reproductive functions until conditions improve for dauer exit. The Roy lab has shown that AMPK is required for germline quiescence and genome integrity during dauer and is specifically required in neurons and the excretory system to regulate these phenotypes. In the absence of AMPK, dauer animals continue to proliferate their germ cells and they lose their germ cell identity. As a result, postdauer adults are sterile. In this manuscript, the authors present evidence the AMPK regulates the RabGAP TBC-7, whose activity regulates the Rab GTPase RAB-7. Inactive RAB-7 is associated with germline quiescence in dauers and postdauer fertility. They show evidence that AMPK phosphorylates TBC-7 to regulate its activity. In addition, the authors show that miR-1 and miR-44 contribute to dauer germline quiescence by playing a role in the downregulation of tbc-7 transcripts. Most interestingly, rescue experiments and tissue-specific RNAi indicate that AMPK, TBC-7, and miR-1/-44 are acting in neurons to regulate the germline during dauer. Future work dissecting the AMPK-dependent signals from the neurons that regulate dauer germline quiescence will be interesting. This manuscript is a significant step in understanding the mechanism of how AMPK regulates dauer phenotypes.

This version of the manuscript has made numerous revisions in response to the previous review. I think that the data presented is compelling, and I do not suggest any additional experiments. However, I do have some suggestions about data interpretation and conclusions which are listed below.

1) The results of the reverse genetic screen in this manuscript are intriguing. To varying degrees, the mutant alleles rescue the postdauer sterility of aak(0) mutants, but they continue to proliferate their germ cells during dauer. Complementation group 3 is particularly interesting in this respect. Since the chromatin marks are largely rescued by the rr166 allele, these data suggest that loss of germ cell identity, and not continued proliferation of germ cells in dauer, is what causes postdauer adult sterility. The rr289 is a great example of this, where the number of germ cells is the same as aak(0) mutants, but 61% of the postdauer adults are fertile. The continued germ cell proliferation may also be contributing to the lack of rescue of lipid storage phenotypes by these alleles, since the stored lipids are likely mobilizing to the germline to support these newly proliferated germ cells. The authors should consider these nuances in interpreting their data, since it points to another AMPK regulated pathway regulating the proliferation of germ cells in dauer.

2) It is not clear to me whether the eggs laid by the aak(0);daf-2;rr166 animals are viable and hatching. Are these animals producing offspring or dead embryos?

3) The tissue specific rescue experiments are done with pan-neuronal promoters, but it would be very interesting to determine which neurons(s) TBC-7 function is required in this context.

4) In the screen to test for Rabs that are acting downstream of TBC-7, it is not clear why the authors only focused on rab-7 when the rab-10 phenotype is almost identical.

5) I agree with the previous reviewer that the rationale for investigating miRNAs in this pathway is not well-justified in the manuscript. MiRNAs and endogenous siRNA pathways are just not the same. I think the manuscript should be changed to reflect the authors’ response to reviewers, such that miRNAs are investigated because “the regulation of tbc-7 by mir-1 is also reported by Nehammer et al eLife 2019 and Gutierrez Perez et al Sci Adv 2021” and that miRNAs are known to regulate dauer phenotypes.

6) I think the authors should be cautious about concluding that AMPK is somehow regulating miRNA production. They do not show evidence that miRNAs miR-1 and miR-44 levels are changing, and the rescue experiment overexpressing miR-1 and miR-44 in Figure 4E is analogous to an RNAi experiment (and is convincing that these miRNAs regulate tbc-7 transcript levels!). If AMPK were to be regulating the activity of miRNA biogenesis proteins, all miRNAs would be affected and it seems likely you would observe more severe phenotypes. A more conservative explanation would be that miR-1/-44 are acting in a parallel pathway to AMPK to regulate TBC-7 function, thus supporting a model whereby miR-1/-44 are acting upstream to regulate TBC-7, and endogenous siRNA pathways are acting downstream to regulate germline chromatin and gene expression in dauer.

Reviewer #2: The authors have addressed my previous comments and those of Reviewer 1 just fine. Ready to publish.

Reviewer #3: This is revised manuscript by Wong et al addressing the discovery of novel suppressors of ampk mutant- induced sterility after dauer, two of which fall in the tbc-7 complementation group. TBC-7 encodes a RabGAP and they expand upon this analysis to identify RABs that contribute to this pathway, and create relevant mutants in rab-7 that are trapped in the active state. They also examine the regulation of TBC-7 and zero in on two miRNAs that appear to regulate tbc-7 based on mutational analysis of the tbc-7 binding sites and overexpression of the miRNAs. They also find that there is a strong AMPK binding site in TBC-7 and create mutants that mimick the phosphorylated and unphosphorylated states, all of which have the expected phenotypes in post-dauer sterility.

Overall the screen is clever and most of the results are substantiated, the manuscript is still very hard to read because of lack of full genotypes in each part of the figure legends in the figures. There are a number of methods that are lacking, making it difficult at times to assess the methodology (when were transgenes vs fosmids used/ what were specific mutations, etc). Further, there are a few cases, where the results are consistent but do not prove a direct interactions, most importantly as to whether AMPK is truly the kinase for TBC-7. Most critical for acceptance is a clarification of these many technical and textual issues. While further experimentation would enhance the paper, softening some of the conclusions might be sufficient.

Specific comments to address prior to publication

• I think it is still confusing to have to read to the end of the figure legend to get the full genotype. I think there would be ways to show that all of the worms have aak or daf-2; aak. Also for the uninitiated, I think it would be helpful in the results to mention that aak(0) = aak-1 aak-2 double mutant.

• All methods should be described in the text itself and should not simply be referenced.

• There is no description of the how the TBC-7::GFP fosmids were constructed, nor any of the transgenes. This information must be included for publication.

• I cannot find a description of the amino acid changed in the RAB-7 GTP locked variant. How was this made and what it the evidence that it has this structure?

• It is unclear in Figure 2D how ANY fertility in the aak(0) germline RNAi is considered not statistically different from complete sterility. If this were identified in the genetic screen, you would have kept it as a putative suppressor. This simply means that the numbers are too low to show significance. I suggest using a Mann Whitney test for this analysis.

• The prior studies showed potential roles for small RNA pathways but technically did not address in which tissues these genes were acting. Rather the previously addressed where AMPK was acting to control germ line chromatin. Thus, I have issue with the statement that “Our previous findings hinted that small RNA homeostasis is misregulated in the germ line during the dauer stage in AMPK mutants,” since there is no evidence either way that the activity was in the germ line. Indeed their attempt to link tbc-7 to a small RNA pathways undermines this statement since they showed earlier in the paper that tbc-7 is required in the neurons!

• Since a prior reviewer mentioned the use of alternative methods to find miRNA targets that also substantiate the examination of miR-1, it seems reasonable to include this as support for their study.

• It would be beneficial to have a more detailed description of the mir-1 mir-44 phenotypes on day 1 and day 7 of dauer since prior studies show that the germ lines do not continually proliferate in dauer. In these mutants do they fail to proliferate early but show delayed divisions?

• Also, the authors state: “Our findings suggest that as the animal spends an extended

414 period of time in the dauer stage, the loss of mir-1 and mir-44 allows for tbc-7 transcripts

415 to be actively expressed.” This is actually not directly test in the paper, but can be substantiated with the WT fosmid tbc-7::GFP that they show in Figure 2.

• It would be nice to the images in Figure 5A presented on a single gel to see if there are differences in mobility that might be expected from a phosphorylated protein.

•There are some issues with initial descriptions of the tbc-7 alleles:

-The description of the complementation analysis of the tbc-7(0) and tbc-7(rr166) fails to show provide one important details: a phenotype. It appears from the text that they were testing viability alone, but this result would also be attained with a crosses to a wild type gene. In the methods, however, they specifically mention having the worms transit through dauer. Where is the data for these results?

-There is no analysis of whether this is a recessive gain-of-function. The argument in the response to reviewers that they cannot make the aak(0); tbc-7 transhets because they are on the same chromosome fails to note that they aak-2 and tbc-7 are 30cM apart and thus, easy to construct genetically.

- Given that rr166 and rr267 are in the same complementation group, it is likely that the mutation in tbc-7 was easier to find because it was likely the only gene with missense mutations in both strains and not the parental. Since EMS causes hundreds of mutations, it is worth commenting that it not so simple as genome sequencing the strain and finding the mutation, as described. In addition, I agree with one of the prior reviewers that at least discussing the nature of rr267 allele here is warranted.

-I am little surprised that the authors did not simply recreate the rr166 mutation by CRISPR. This would have eliminated any concerns about background mutations.

-A prior reviewer asked for a summary of the post-dauer somatic effects to be included. Considering that the authors mention that rr166 suppresses most of the somatic defects, including a sentence about the phenotypes is warranted, and a reference to the prior paper. This is important given the statements in the following paragraph (line 184-186) about rr166 specificity.

–In summary of this section about the alleles: the authors should tighten up the complementation analysis and provide information on the nature of the rr267 allele and reiterate the non-germ line phenotypes that are assayed for publication.

•The use of phosphomimetic and deficient mutants is convincing, but true proof for direct phosphorylation is lacking and would come from either in vitro studies or creation of a phosphor specific antibody.

• It is still unclear why rab-7 and not rab-5, for example, was chosen for further analysis. The authors state that “..when rab-7 was compromised in the aak(0); tbc-7 mutants, the post-dauer fertility was reduced to a level similar to that of post-dauer aak(0) mutants.” However the aak(0) mutant in Fig 1 and S1 how 0% fertility, like rab-5, and not 4% like rab-7 in the aak(0) tbc-7

• Why was rab-5 omitted from Figure 3B?

• The conclusion that “its expression in the neurons must be targeted by AMPK during the onset of the dauer stage to instruct. the germ line to execute quiescence” is not supported by the data at this point in the paper, at least. There is as yet no evidence that TBC-7 works downstream or is a target of AMPK, rather the data thus far simply supports that it can suppress loss of AMPK function.

Minor technical details that should addressed

Line 282, you should mention how big the upstream and downstream regulatory regions are in the fosmid.

Line 85: Paradis and Ruvkun does not address oocyte development and fertility

Line 323-325: I think it is worth saying that the activity of TBC-7 is inferred since technically you have not measured the activity.

daf-2; aak(0); tbc-7 since aak(0) is on the X as is tbc-7, the genotype should be written daf-2; aak(0) tbc-7 or daf-2; aak(0), tbc-7 (though this is often confusing and not preferred)

In figure 3B, the control genotype should be listed in the key. It was unclear, for example that this was the data to support the statement, “Indeed, rab-7 RNAi during the dauer stage caused post-dauer sterility.” (lines 335-336)? Both the figure was not reference and it was unclear the figure was the rab-7(RNAi) without daf-2;aak(0);tbc-7.

Mutagenesis Methods:

What does F1 animals were isolated to …. does that mean that they were individually plated?

Am I reading correctly that the F2 worms were only at 25°C for 48 hours. So they never actually molted into dauer, but went into L2d and then were recovered

The authors should include a caveat to their Western blot analysis of TBC-7 in AMPK mutants that although the levels do not change, they cannot rule out that protein levels are not changed only in neurons, where both proteins appear to act together to control germline quiescence.

Textual issues that should be addressed

Line 82: “Both starved C. elegans, or…” no comma. I also do not understand this statement since daf-2 mutants are fertile. I think you are trying to discuss what happened during dauer, but this is totally unclear.

The paragraph in the introduction (Line 114 -126) is confusing. It seems to be setting up a non-germ cell autonomous role for AMPK but then Line 128 is not “surprising” but rather expected since we already know it can act in other tissues. The idea that the sensing can be in neurons could regulate the germ line is reasonable, but the rest of the paragraph is non-specific about which tissues AMPK is in.

Lines 175-178: These ideas are not intuitively connected: “Moreover, rr166 also suppressed most of the post-dauer somatic defects, suggesting either that this allele is not unique to the germ line, or that it affects the somatic tissues to adjust cell division timing for the duration of the diapause through its effect on the germ cells.” rr66 can have a somatic function independent of its role in dauer.

Line 257: tbc-7 lethality needs a reference

Line 261 should read “indicated that it is indeed a”

Line 287 -288 “Because of the claim that TBC-7 might also function in the body wall muscles during the dauer stage”. should be referenced

Lines 288, 291, 400 and 683, TBC-7 should be the gene and not the protein.

Line 297 “the post-dauer sterility and the germline hyperplasia is” should be “are”

Line 304 “germ cell” should be “germ line”

Line 302 -306. Run on.

Supplemental Line 86 should be TBC-7::GFP

The term “germ cell integrity” usually references to genomic integrity not proliferative status or developmental stage. I think state, germ cell proliferation or differentiation would be more appropriate here.

Lines 373-378 are very confusing since it is not always clear when the authors state “misregulated TBD-7” that they are referring to the daf-2; aak(0) background. Further and again in this section, as earlier in the text, they have not yet proven the TBD-7 is misregulated, per se, in this background, rather they infer it from the phenotypes. The text would be best to stick to the genetic terms for these phenotypes.

Line 403-405 would benefit from precise genotypic descriptions

Line 448: this needs a figure reference

Line 451 should state in neurons since the rgef promoter is used for this analysis.

Line 500 phosphomimetic spelled incorrectly

Line 509-510 is a repeat of 468-470.

Line 554: fix “but not it is not”

Intertissular does not appeat to be a word and should be replaced with “intertissue”

Line 705 “were transited” should be “were made to transit”

Line 707 should read “to determine if…”

Reviewer #4: Starting from an unbiased genetic screen for suppressors of certain phenotypes associated with loss of AMPK, the manuscript details a role for the RabGAP tbc-7 in neurons. The supposition that tbc-7 is required in neurons is based on heterologous gene expression and tissue-modulated RNAi, as well as testing and ruling out other tissue over-expression (muscle, excretory cell) that were previously identified as sources of non-autonomous AMPK effects. An RNAi survey of RABs identifies a particularly relevant target, RAB-7, that phenocopies loss of AMPK when mutated and partially suppresses loss of AMPK when locked in the GTP-bound active state. Further, TBC-7 bears potential AMPK substrate sequences that appear functional based on substitution with non-phosphorylatable or phosphomimetic amino acids. Based on sequence analysis of the tbc-7, 3’UTR sequence, a role for mir-1 and mir-44 is also explored, revealing regulation important after a long dauer duration.

The results are novel and of general interest to many, including those interested in diapause, AMPK function, germline development, and cellular quiescence. In general, the conclusions are supported by the experiments (however, see below). Further, the work and the thoughtful discussion open exciting questions about how RABs regulate signaling from neurons to the germ line.

Comments and suggestions:

1. Screen, Table 1 and identification of tbc-7:

a. Table 1 – Are these averages? What is N? Explain “wild type-like” designation. Why does “no. of germ cells” for aak(0) differ between Table 1 and Figure 1? A table legend would be helpful.

b. Which specific amino acids are mutated in the two tbc-7 alleles (this information is available from the WGS, so maybe just include it)? Do the specific changes lend any suggestion as to how they affect the GAP function? Does molecular change in rr267 suggest why this allele may be a less efficient suppressor?

c. While the best proof for rr166 being the causal suppressor mutation would be to reintroduce the same mutation into the starting background via CRISPR/Cas9 gene editing, the complementation rescue with the fosmid together with the RNAi phenotype are pretty convincing. Nevertheless, the authors may wish to include in Figure 1 data with both the rr166 and rr267 alleles. In particular, Figure 1D rr166 strain is quite remarkably identical to the aak(0) and would be bolstered by a similar pattern of effects on phenotypes shown in Fig 1 A-C but not D.

2. Figure 1 (and other figures):

a. To aid the non-cognoscenti, I also suggest more descriptive Y axes in Figure 1 and elsewhere in the manuscript. Consider (A) Post-dauer % of egg-laying animals, (B) Dauer germ cell number. Also consider putting full relevant genotypes in the figures and strain names in the figure legends. At least indicate on the figure that all strains also carry daf-2(e1370)? Figures in previous papers from this lab include necessary genotypes. It is very confusing to figure out what is being tested without combing through the figure legends.

b. Often n is indicated as “n=50”. Are all of these in one trial? Please state for each figure.

3. Fig. 4: Reword the Figure title since the function of mir-1 and mir-44 was not directly tested in neurons, nor was the expression of tbc-7 directly tested.

4. Methods: Methods are insufficient and, in addition to specifics below, require editing with attention to italics and other nomenclature.

a. Screen – please make screen strategy more accessible to reader. What strain (name + genotype) was mutagenized? Exactly what phenotype was screened for (presumably post-dauer % egg-laying adults?)? Provide more details on WGS analysis.

b. Complementation – the text requires at least a reference for “complementation analysis was conducted as previously described following injection of myo-2…”. Ideally more details on the complementation analysis would be provided.

c. Microscopy – indicate how ratio quantification was performed.

d. Site-directed mutagenesis – include all methods on how 3’ UTR deletions and amino acid substitutions were made, including primers, etc.

e. Transgenes – state exactly what is in the transgenes driven by unc-199, myo-3 and sulp-5 promoters (tbc-7 cDNA?);

f. Strain list – include strains bearing all transgenes (including 3’ UTR, phosphor site mutations, and heterologous expression) in the strain list together worm strain names and plasmid names, and references where necessary.

5. Supplement Tables

a. Table S2 & S4: Legend and/or headings could be improved to help reader understand the set up – table is confusing.

b. Table S3: Also has daf-2 in the background? If so, state.

6. Other:

a. Line 412-413, consider restating. That mir-1 and -44 mutants enhance post-dauer sterility of rab-7 RNAi does not necessarily indicate a linear pathway.

b. Line 543, restate. The screen was perhaps near saturation, but not “indeed saturated” since Groups 4 and 5 are represented by only one recessive allele each.

c. Line 558. Reference to Kadekar and Roy 2019 – This paper (Fig 8) shows that AMPK expression in neurons and excretory cell can rescue post-dauer egg-laying and germline hyperplasia during dauer. However, I do not see results on premature lethality during dauer (perhaps I missed?).

d. Lines 707-712. Alert the reader to additional information in S4 Figure.

**Have all data underlying the figures and results presented in the manuscript been provided?**

Reviewer #1: **No: **Not all of the raw data for independent trials have been included.

Reviewer #2: Yes

Reviewer #3: **No: **Deetailed methods are lacking.

Reviewer #4: Yes

PLOS authors have the option to publish the peer review history of their article (what does this mean?). If published, this will include your full peer review and any attached files.

Reviewer #1: No

Reviewer #2: No

Reviewer #3: No

Reviewer #4: No

---

## [Editor Report · Decision Letter 1]

23 Mar 2023

Dear Dr Roy,

We are pleased to inform you that your manuscript entitled "Germline stem cell integrity and quiescence are controlled by an AMPK-dependent neuronal trafficking pathway" has been editorially accepted for publication in PLOS Genetics. Congratulations!

The revised manuscript was evaluated and reviewed by members of the editorial board; overall, there is a consensus that the concerns raised during the previous round of reviews (and, before that, at review commons) have been addressed.

Yours sincerely,

Gregory S. Barsh

Editor-in-Chief

PLOS Genetics

Gregory Copenhaver

Editor-in-Chief

PLOS Genetics

Comments from the reviewers (if applicable):

**Data Deposition**

http://datadryad.org/submit?journalID=pgenetics&manu=PGENETICS-D-22-01165R1

**Press Queries**

---

## [Editor Report · Acceptance letter]

10 Apr 2023

PGENETICS-D-22-01165R1 

Germline stem cell integrity and quiescence are controlled by an AMPK-dependent neuronal trafficking pathway 

Dear Dr Roy, 

We are pleased to inform you that your manuscript entitled "Germline stem cell integrity and quiescence are controlled by an AMPK-dependent neuronal trafficking pathway" has been formally accepted for publication in PLOS Genetics! Your manuscript is now with our production department and you will be notified of the publication date in due course.

With kind regards,

Zsofi Zombor

PLOS Genetics

On behalf of:
